

# Snowfall in the Alps: Evaluation and projections based on the EURO-CORDEX regional climate models

Prisco Frei [1], Sven Kotlarski [2,*], Mark A. Liniger [2], Christoph Schär [1]

[1] Institute for Atmospheric and Climate Sciences, ETH Zurich, 8006, Zurich, Switzerland
[2] Federal Office of Meteorology and Climatology, MeteoSwiss, 8058 Zurich-Airport, Switzerland

[*] Corresponding author: *sven.kotlarski@meteoswiss.ch*

**Abstract.** Twenty-first century snowfall changes over the European Alps are assessed based on high-resolution regional climate model (RCM) data made available through the EURO-CORDEX initiative. Fourteen different combinations of global and regional climate models with a target resolution of 12 km, and two different emission scenarios are considered. A newly developed method to separate snowfall from total precipitation based on near-surface temperature conditions and accounting for subgrid topographic variability is employed. The evaluation of the simulated snowfall amounts against an observation-based reference indicates the ability of RCMs to capture the main characteristics of the snowfall seasonal cycle and its elevation dependency, but also reveals considerable positive biases especially at high elevations. These biases can partly be removed by the application of a dedicated RCM bias correction that separately considers temperature and precipitation biases.

Snowfall projections reveal a robust signal of decreasing snowfall amounts over most parts of the Alps for both emission scenarios. Domain and multimodel-mean decreases of mean September-May snowfall by the end of the century amount to -25% and -45% for RCP4.5 and RCP8.5, respectively. Snowfall in low-lying areas in the Alpine forelands could be reduced by more than -80%. These decreases are driven by the projected warming and are strongly connected to an important decrease of snowfall frequency and snowfall fraction and are also apparent for heavy snowfall events. In contrast, high-elevation regions could experience slight snowfall increases in mid-winter for both emission scenarios despite the general decrease of the snowfall fraction. These increases in mean and heavy snowfall can be explained by a general increase of winter precipitation and by the fact that, with increasing temperatures, climatologically cold areas are shifted into a temperature interval which favours higher snowfall intensities.



## 1 Introduction

Snow is an important resource for the Alpine regions, be it for tourism, hydropower generation, or water management (Abegg et al., 2007). According to the Swiss Federal Office of Energy (SFOE) hydropower generation accounts for approximately 55% of the Swiss electricity production (SFOE, 2014). Consideration of changes in snow climatology needs to address aspects of both snow cover and snow fall. In the recent past, an important decrease of the mean snow cover depth and duration in the Alps was observed (e.g, Laternser and Schneebeli, 2003; Marty, 2008; Scherrer et al., 2004). Future projections using climate model simulations of the anthropogenic greenhouse effect indicate a further substantial reduction (Schmucki et al., 2015a; Steger et al., 2013), strongly linked to the expected rise of temperatures (e.g., CH2011, 2011; Gobiet et al., 2014). On regional and local scales rising temperatures exert a direct influence on snow cover in two ways: First, total snowfall sums are expected to decrease by a decreasing probability for precipitation to fall as snow and a decreasing snowfall fraction (ratio between solid and total precipitation). Second, snow on the ground is subject to faster and accelerated melt. These warming-induced trends might be modulated by changes in atmospheric circulation statistics.

Although the snowfall fraction is expected to decrease at lower elevations during the 21st century, extraordinary snowfall events can still leave a trail of destruction. A recent example was the winter 2013/2014 with record-breaking heavy snowfall events along the southern rim of the European Alps (e.g., Techel et al., 2015). The catastrophic effects of heavy snowfall range from avalanches and floods to road or rail damage. In extreme cases these events can even result in the weight-driven collapse of buildings or loss of human life (Marty and Blanchet, 2011). Also mean snowfall conditions, such as the mean number of snowfall days in a given period, can be of high relevance for road management (e.g. Zubler et al., 2015) or airport operation. Projections of future changes in the snowfall climate, including mean and extreme conditions, are therefore highly relevant for long-term planning and adaptation purposes in order to assess and prevent related socio-economic impacts and costs.

21$^{st}$ century climate projections typically rely on climate models. For large-scale projections, global climate models (GCMs) with a rather coarse spatial resolution of 100 km or more are used. For assessing regional to local scale impacts, where typically a much higher spatial resolution of the projections is required, a GCM can be dynamically downscaled by nesting a regional climate model (RCM) over the specific domain of interest (Giorgi, 1990). In such a setup, the GCM provides the lateral boundary conditions to the RCM. One advantage of climate models is the ability to estimate climate change in a physically based manner under different greenhouse gas (GHG) emission scenarios. With the Intergovernmental Panel on Climate Change's (IPCC) release of the Fifth Assessment Report (AR5; IPCC, 2013) the so-called representative concentration pathway (RCP) scenarios have been introduced  (Moss et al., 2010) which specify GHG concentrations and corresponding emission pathways for several radiative forcing targets. To estimate inherent projection uncertainties, ensemble approaches employing different climate models, different greenhouse gas scenarios, and/or different initial conditions are being used (e.g., Deser et al., 2012; Hawkins and Sutton, 2009; Rummukainen, 2010).



Within the last few years several studies targeting the future global and European snowfall evolution based on climate model ensembles were carried out (e.g., de Vries et al., 2013; de Vries et al., 2014; Krasting et al., 2013; O'Gorman, 2014; Piazza et al., 2014; Räisänen, 2016; Soncini and Bocchiola, 2011). Most of these analyses are based on GCM output or older generations of RCM ensembles at comparatively low spatial resolution, which are not able to properly resolve snowfall events over regions with complex topography. New generations of high resolution RCMs are a first step toward an improvement on this issue. This is in particular true for the most recent high-resolution regional climate change scenarios produced by the global CORDEX initiative (Giorgi et al., 2009) and its European branch EURO-CORDEX (Jacob et al., 2014). The present work aims to exploit this recently established RCM archive with respect to future snowfall conditions over the area of the European Alps. By covering both model evaluation and high-resolution future snowfall projections we are addressing the following main objectives:

**Snowfall separation on a (coarse resolution) RCM grid.** Raw snowfall outputs are not available for all members of the EURO-CORDEX RCM ensemble and, furthermore, a gridded observational snowfall product that could serve as reference for RCM evaluation does not exist. Therefore, an adequate snowfall separation technique, i.e., the derivation of snowfall amounts based on readily available daily near-surface temperature and precipitation data, is required. Furthermore, as the observational and simulated grids of the two latter variables are typically not available at the same horizontal resolution, we seek for a snowfall separation method that accounts for the topographic subgrid variability of snowfall on the coarser (RCM) grid.

**Snowfall bias correction.** Even the latest generation of RCMs is known to suffer from systematic model biases (e.g., Kotlarski et al., 2014). In GCM-driven setups as employed within the present work these might partly be inherited from the driving GCM. To remove such systematic model biases in temperature and precipitation, a simple bias correction methodology will be developed and employed in the present work. To assess its performance and applicability, different snowfall indices in the bias-corrected and not bias-corrected output will be compared against observational estimates.

**Snowfall projections for the late 21st century.** Climate change signals for various snowfall indices over the Alpine domain and for specific elevation intervals, derived by a comparison of 30-year control and scenario periods, will be analysed under the assumption of the RCP8.5 emission scenario. In addition, we aim to identify and quantify the main drivers of future snowfall changes and, in order to assess emission scenario uncertainties, compare RCP8.5-based results with experiments assuming the more moderate RCP4.5 emission scenario.

On centennial time scales, two main drivers of future snowfall changes over the European Alps with competing effects on snowfall amounts are apparent: (1) Mean winter precipitation is expected to increase over most parts of the European Alps and in most EURO-CORDEX experiments (e.g., Rajczak et al., in prep.; Smiatek et al., 2016) which in principle could lead to higher snowfall amounts. (2) Temperatures are projected to considerably rise throughout the annual cycle (e.g., Gobiet et al., 2014; Smiatek et al., 2016; Steger et al., 2013) with the general effect of a decreasing snowfall frequency and fraction, thus potentially leading to a reduction in overall snowfall amounts changes.





Separating the above two competing factors is one of the targets of the current study. A potential complication is that changes in daily precipitation frequency (here events > 1 mm/day) and precipitation intensity (average amount on wet days) can change in a counteracting manner (e.g., Fischer et al., 2015; Rajczak et al., 2013), and that relative changes are not uniform across the event category (e.g. Ban et al., 2015; Fischer and Knutti, 2016).

The article is structured as follows: Section 2 describes the data used and methods employed. In Sections 3 and 4 results of the bias correction approach and snowfall projections for the late 21st century are shown, respectively. The latter are further discussed in Section 5 while overall conclusions and a brief outlook are provided in Section 6. Additional supporting figures are provided in the supplementary material (prefix 'S' in Figure numbers).

## 2 Data and methods

### 2.1 Observational data

To estimate observation-based snowfall, two gridded data sets, one for precipitation and one for temperature, derived from station observations and covering the area of Switzerland are used. Both data sets are available on a daily basis with a horizontal resolution of 2 km for the entire evaluation period 1971-2005 (see Sec. 2.3).

The gridded precipitation data set (RhiresD) represents a daily analysis based on a high-resolution rain-gauge network (MeteoSwiss, 2013a) which has a balanced distribution in the horizontal but under-represents high altitudes (Frei and Schär, 1998; Isotta et al., 2014; Konzelmann et al., 2007). Albeit the data set's resolution of 2 km, the effective grid resolution as represented by the mean inter-station distance is about 15 - 20 km and thus comparable to the available climate model data (see Sec. 2.2). The dataset has not been corrected for the systematic measurement bias of rain gauges (e.g., Neff, 1977; Sevruk, 1985; Yang et al., 1999).

The gridded near-surface air temperature (from now on simply referred to as *temperature*) data set (TabsD) utilises a set of homogeneous long-term station series (MeteoSwiss, 2013b). Despite the high quality of the underlying station series, errors might be introduced by unresolved scales and interpolation uncertainty (Frei, 2014). The unresolved effects of land cover or local topography, for instance, probably lead to an underestimation of spatial variability. Another problem arises in inner Alpine valleys, where the presence of cold air pools is systematically overestimated.

### 2.2 Climate model data

In terms of climate model data we exploit a recent ensemble of regional climate projections made available by EURO-CORDEX (www.euro-cordex.net), the European branch of the World Climate Research Programme's CORDEX initiative (www.cordex.org; Giorgi et al., 2009). RCM simulations for the European domain were run at a resolution of 50 km (EUR-44) and 12.5 km (EUR-11) with both re-analysis boundary forcing (Kotlarski et al., 2014; Vautard et al., 2013) and GCM-forcing (Jacob et al., 2014). The latter include historical control simulations and future projections based on RCP



greenhouse gas and aerosol emission scenarios. Within the present work we employ all GCM-driven EUR-11 simulations for which control, RCP4.5 and RCP8.5 runs are currently available. This yields a total set of 14 GCM-RCM model chains, combining five driving GCMs with seven different RCMs (Tab. 1). We exclusively focus on the higher resolved EUR-11 simulations and disregard the coarser EUR-44 ensemble due to the apparent added value of the EUR-11 ensemble with respect to regional-scale climate features in the complex topographic setting of the European Alps (e.g., Giorgi et al., 2016; Torma et al., 2015).

It is important to note that each of the six RCMs considered uses an individual grid cell topography field. Model topographies for a given grid cell might therefore considerably differ from each other, and also from the observation-based orography. Hence, it is not meaningful to compare snowfall values at individual grid cells since the latter might be situated at different elevations. Therefore, most analyses of the present work were carried out as a function of elevation, i.e., by averaging climatic features over distinct elevation intervals.

### 2.3 Analysis domain and periods

The arc-shaped European Alps - with a West-East extent of roughly 1200 km , a total of area 190'000 $km^2$ and a peak elevation of 4810 m a.s.l. (Mont Blanc) - are the highest and most prominent mountain range which is entirely situated in Europe. In the present work, two different analysis domains are used. The evaluation of the bias correction approach depends on the observational data sets RhiresD and TabsD (see Sec. 2.1). As these cover Switzerland only, the evaluation part of the study (Sec. 3) is constrained to the Swiss domain (Fig. 1, bold line). For the analysis of projected changes of different snowfall indices (Sec. 4 and 5) a larger domain covering the entire Alpine crest with its forelands is considered (Fig. 1, coloured region).

Our analysis is based on three different time intervals. The evaluation period (EVAL) 1971-2005 was used for the calibration and validation of the bias correction approach. Future changes of snowfall indices were computed by comparing a present day control period (1981-2010, CTRL) to a future scenario period at the end of the 21$^{st}$ century (2070-2099, SCEN). For all periods (EVAL, CTRL and SCEN), the summer months June, July and August (JJA) are excluded from any statistical analysis. In addition to seasonal mean snowfall conditions, i.e., averages over the nine-month period from September to May, we also analyse the seasonal cycle of individual snowfall indices at monthly resolution.

### 2.4 Analysed snowfall indices and change signals

A set of six different snowfall indices is considered (Tab. 2). Mean snowfall ($S_{mean}$) refers to the (spatio-) temporally-averaged snowfall amount in mm SWE (note that from this point on we will use the term "mm" as a synonym for "mm SWE" as unit of several snowfall indices). The two indices heavy snowfall ($S_{q99}$) and maximum 1-day snowfall ($S_{1d}$) allow the assessment of projected changes in heavy snowfall events and amounts. $S_{1d}$ is derived by averaging maximum 1-day snowfall amounts over all individual months/seasons of a given time period (i.e., by averaging 30 maximum values in the case of the CTRL and SCEN period), while $S_{q99}$ is calculated from the grid point-based 99$^{th}$ all-day snowfall





percentile of the daily probability density function (PDF) for the entire time period considered. We use all-day percentiles as the use of wet-day percentiles leads to conditional statements that are often misleading (see the analysis in Schär et al. 2016). Note that the underlying number of days differs for seasonal (September-May) and monthly analyses. Snowfall frequency ($S_{freq}$) and mean snowfall intensity ($S_{int}$) are based on a wet-day threshold of 1 mm/day and provide additional information about the distribution and magnitude of snowfall events, while the snowfall fraction ($S_{frac}$) describes the ratio of solid precipitation to total precipitation. As climate models tend to suffer from too high occurrence of drizzle and as small precipitation amounts are difficult to measure, daily precipitation values smaller or equal to 0.1 mm were initially set to zero in both the observations and the simulations.

Projections are assessed by calculating two different types of changes between the CTRL and the SCEN period. The absolute change signal ($\Delta$) of a particular snowfall index X (see Tab.2)

$$\Delta X = X_{SCEN} - X_{CTRL} \tag{1}$$

and the relative change signal ($\delta$) which describes the change of the snowfall index as a percentage of its CTRL period value

$$\delta X = \left(\frac{X_{SCEN}}{X_{CTRL}} - 1\right) \cdot 100 \tag{2}$$

To prevent erroneous data interpretation due to possible large relative changes of small CTRL values, certain grid boxes were masked out before calculating and averaging the signal of change. This filtering was done by setting threshold values for individual indices and statistics (see Table 2 ).

**2.5 Separating snowfall from total precipitation**

Due to (a) the lack of a gridded observational snowfall data set and (b) the fact that not all RCM simulations available through EURO-CORDEX provide raw snowfall as an output variable, a method to separate solid from total precipitation depending on near-surface temperature conditions is developed. This method also allows for a more physically-based bias correction of simulated snowfall amounts (see Sec. 2.6). Due to the temperature dependency of snowfall occurrence, snowfall biases of a given climate model cannot be expected to remain constant under current and future (i.e., warmer) climate conditions. For instance, a climate model with a given temperature bias might pass the snow-rain temperature threshold earlier or later than reality during the general warming process. Hence, traditional bias correction approaches based only on a comparison of observed and simulated snowfall amounts in the historical climate would possibly fail due to a non-stationary bias structure.

The simplest approach to separate snowfall from total precipitation is to fractionate the two phases binary by applying a constant snow fractionation temperature (e.g., de Vries et al., 2014; Schmucki et al., 2015a; Zubler et al., 2014). More sophisticated methods estimate the snow fraction $f_s$ dependence on air temperature with linear or logistic relations (e.g., Kienzle, 2008; McAfee et al., 2014). In our case, the different horizontal resolutions of the observational (high resolution of 2 km) and simulated (coarser resolution of 12 km) data sets further complicate a proper comparison of the respective snowfall amounts. Thus, we explicitly analysed the snowfall amount dependency on the grid resolution





and exploited possibilities for including subgrid-scale variability in snowfall separation based on coarse
grid information. This approach is important as especially in Alpine terrain a strong subgrid variability
of near-surface temperatures due to orographic variability has to be expected, with corresponding
effects on the subgrid snowfall fraction.
For this preparatory analysis, which is entirely based on observational data, a reference snowfall is
derived. It is based on the approximation of snowfall by application of a fixed temperature threshold to
daily total precipitation amounts on the high resolution observational grid (2 km) and will be termed
*Subgrid method* thereafter: First, the daily snowfall $S'$ at each grid point of the observational data set at
high resolution (2 km) is derived by applying a snow fractionation temperature $T^*$=2°C. The whole
daily precipitation amount $P'$ is accounted for as snow $S'$ (i.e., $f_s$=100%) for days with daily mean
temperature $T' \leq T^*$. For days with $T' > T^*$, $S'$ is set to zero and $P'$ is attributed as rain (i.e., $f_s$=0%). This
threshold approach with a fractionation temperature of 2°C corresponds to the one applied in previous
works and results appear to be in good agreement with station-based snowfall measurements (e.g.,
Zubler et al., 2014). The coarse grid (12 km) reference snowfall $S_{SG}$ is determined by averaging the
sum of separated daily high resolution $S'$ over all $n$ high-resolution grid points $i$ located within a specific
coarse grid point $k$. I.e., at each coarse grid point $k$
$$S_{SG} = \frac{1}{n} \cdot \sum_{i=1}^{n} P'_i \left[ T'_i \leq T^* \right] = \frac{1}{n} \sum_{i=1}^{n} S'_i \qquad (3)$$
For comparison, the same binary fractionation method with a temperature threshold of $T^*$=2°C is
directly applied on the coarse 12 km grid (*Binary method*). For this purpose, total precipitation $P'$ and
daily mean temperature $T'$ of the high-resolution data are conservatively remapped to the coarse grid
leading to $P$ and $T$, respectively. Compared to the *Subgrid method*, the *Binary method* neglects any
subgrid variability of the snowfall fraction. As a result, the *Binary method* underestimates $S_{mean}$ and
overestimates $S_{q99}$ for all elevation intervals (Fig. 2). The underestimation of $S_{mean}$ can be explained by
the fact that even for coarse grid temperature above $T^*$ individual high-elevation subgrid cells (at which
$T' \leq T^*$) can receive substantial snowfall amounts, a process that is not accounted for by the *Binary*
*method*. Furthermore, following O'Gorman (2014), heavy snowfall events are expected to occur in a
narrow temperature range below the rain-snow transition. As the *Binary method* in these temperature
ranges always leads to a snowfall fraction of 100%, too large $S_{q99}$ values would result.
To take into account these subgrid effects, a more sophisticated approach – referred to as the
*Richards method* – is developed here. This method is based upon a generalised logistic regression
(Richards, 1959).  Here, we apply this regression to relate the surface temperature $T$ to the snow
fraction $f_s$ by accounting for the topographic subgrid variability. At each coarse grid-point $k$, the
*Richards method*-based snowfall fraction $f_{s,Rl}$ for a given day is hence computed as follows:
$$f_{s,RI}(T_k) = \frac{1}{\left[ 1 + C_k \cdot e^{D_k \cdot (T_k - T^*)} \right]^{\frac{1}{C_k}}} \qquad (4)$$
with $C$ as the point of inflexion, and the growth rate $D$. $T_k$ is the daily mean temperature of the
corresponding coarse grid box $k$ and $T^*$=2°C the snow fractionation temperature. First, we estimate
the two parameters $C$ and $D$ of Equation 4 for each single coarse grid point $k$ by minimizing the least-



square distance to the $f_s$ values derived by the *Subgrid method* via the reference snowfall $S_{SG}$ (local
fit). Second, C and D are expressed as a function of the topographic standard deviation $\sigma_h$ of the
corresponding coarse resolution grid point only (Fig. S1; global fit). This makes it possible to define
empirical functions for both *C* and *D* that can be used for all grid points *k* in the Alpine domain and that
depend on $\sigma_h$ only.
$$\sigma_{h,k} = \sqrt{\frac{\sum_i^n (h_i - \overline{h_k})^2}{n-1}} \tag{5}$$

$$C_k = \frac{1}{(E - \sigma_{h,k} \cdot F)} \tag{6}$$

$$D_k = G \cdot \sigma_{h,k}^{-H} \tag{7}$$

Through a minimisation of the least square differences the constant parameters in Equations 6 and 7
are calibrated over the domain of Switzerland and using daily data from the period September to May
1971-2005 leading to values of $E$=1.148336, $F$=0.000966 m$^{-1}$, $G$=143.84113 °C$^{-1}$ and $H$=0.8769335.
Note that $\sigma_h$ is sensitive to the resolution of the two grids to be compared (cf. Eq. 5). It is a measure for
the uniformity of the underlying topography. Small values indicate a low subgrid topographic variability,
such as in the Swiss low-lands, while high values result from non-uniform elevation distributions, such
as in areas of inner Alpine valleys. Figure S1 (panel c) provides an example of the relation between
daily mean temperature and daily snow fraction $f_s$ for grid cells with topographical standard deviations
of 50 m and 500 m, respectively. The snowfall amount $S_{RI}$ for a particular day and a particular coarse
grid box is finally obtained by multiplying the corresponding $f_{s,RI}$ and $P$ values. A comparison with the
*Subgrid method* yields very similar results. For both indices $S_{mean}$ and $S_{q99,}$ mean ratios across all
elevation intervals are close to 1 (Fig. 2). At single grid points, maximum deviations are not larger than
1±0.1. Note that for this comparison calibration and validation period are identical (EVAL period).
Based on this analysis, it has been decided to separate snowfall according to the *Richards method*
throughout this work in both the observations and in the RCMs. The observation-based snowfall
estimate obtained by applying the *Richards method* to the observational temperature and precipitation
grids after spatial aggregation to the 0.11° RCM resolution will serve as reference for the RCM bias
correction and will be termed *reference* hereafter. One needs to bear in mind that the parameters $C$
and $D$ of the Richards method were fitted for the Swiss domain only and were later on applied to the
entire Alpine domain (cf. Fig. 1).

### 2.6 Bias correction approach

Previous work has revealed partly substantial temperature and precipitation biases of the EURO-
CORDEX RCMs over the Alps (e.g. Kotlarski et al., 2014; Smiatek et al., 2016), and one has to expect
that the separated snowfall amounts are biased too. This would especially hamper the interpretation of
absolute climate change signals of the considered snow indices. We explore possibilities to bias-
correct the simulated snowfall amounts and to directly integrate this bias correction into the snowfall
separation framework of Section 2.5. We compare results with and without employment of the bias
correction procedure outlined below. A simple two-step approach that separately accounts for
precipitation and temperature biases and their respective influence on snowfall is chosen. The bias



correction is calibrated in the EVAL period for each individual GCM-RCM chain and over the region of
Switzerland, and is then applied to both the CTRL and SCEN period of each chain and for the entire
Alpine domain. To be consistent in terms of horizontal grid spacing, the observational data sets
RhiresD and TabsD (see Sec. 2.1) are conservatively regridded to the RCM resolution beforehand.
In a first step, total simulated precipitation was adjusted by introducing an elevation-dependent
correction factor which corrects for precipitation biases regardless of temperature. For this purpose,
mean precipitation ratios (RCM simulation divided by observational analysis) for 250 m elevation
intervals were calculated (Fig. S2). An almost linear relationship of these ratios with elevation was
found. Thus, a linear regression between the intervals from 250 m a.s.l. to 2750 m a.s.l. was used for
each model chain separately to estimate a robust correction factor. As the number of both RCM grid
points and measurement stations at very high elevations >2750 m is small (see Sec. 2.1) and biases
are subject to a considerable sampling uncertainty, these elevations were not considered in the
regression. Overall the fits are surprisingly precise except for the altitude bins above 2000 m (Fig. S2).
The precipitation adjustment factors ($P_{AF}$) for a given elevation were then obtained as the inverse of
the fitted precipitation ratios. Multiplying simulated precipitation $P$ with $P_{AF}$ for the respective model
chain and elevation results in the corrected precipitation:
$$P_{corr} = P \cdot P_{AF} \tag{8}$$
For a given GCM-RCM chain and for each elevation interval, the spatially and temporally averaged
corrected total precipitation $P_{corr}$ approximately corresponds to the observation-based estimate.
In a second step of the bias correction procedure, temperature biases are accounted for. For this
purpose the initial snow fractionation temperature T*=2°C of the Richards separation method (see Sec
2.5) is shifted to the value $T_a^*$ for which the spatially and temporally averaged simulated snowfall
amounts for elevations below 2750 m a.s.l. match the respective observation-based reference (see
above). Compared to the adjustment of total precipitation, $T_a^*$ is chosen independent of elevation, but
separately for each GCM-RCM chain. After this second step of the bias correction, the spatially (Swiss
domain) and temporally (September to May) averaged simulated snowfall amounts below 2750 m by
definition match the reference. Hence, the employed simple bias correction procedure corrects
domain-mean snowfall biases averaged over the entire season from September to May. It does,
however, not correct for biases in the spatial snowfall pattern, in the seasonal cycle, or in the temporal
distribution of daily values. Note that, as the underlying high-resolution data sets are available over
Switzerland only, the calibration of the bias correction methodology is correspondingly restricted, but
the correction is then applied to the whole Alpine domain. This approach is justified as elevation-
dependent mean winter precipitation and temperature biases of the RCMs employed – assessed by
comparison against the coarser-resolved EOBS reference dataset (Haylock et al., 2008) -  are very
similar over Switzerland and over the entire Alpine analysis domain (Figs. S3 and S4).



## 3 Evaluation

### 3.1 RCM raw snowfall

We first carry out an illustrative comparison of RCM raw snowfall amounts (for those simulations only that directly provide snowfall flux) against station observations of snowfall, in order to determine whether the simulated RCM snowfall climate contains valid information despite systematic biases. To this end, simulated raw snowfall amounts of nine EURO-CORDEX simulations (see Tab. 1) averaged over 250 m-elevation intervals in the range 950 – 1650 m are compared against observations derived from measured fresh snow sums from 29 MeteoSwiss stations with data available for at least 80% of the EVAL period. For this purpose a mean snow density of 100 kg/m$^3$ for the conversion from measured snow height to water equivalent is assumed. Note that this simple validation does not explicitly correct for the scale gap between grid-cell based RCM output and single-site observations. At low elevations simulated mean September-May raw snowfall sums match the observations well while differences are larger aloft (Fig. 3a). The positive bias at high elevations might arise from the fact that (the very few) observations were made at a specific location while simulated grid point values of the corresponding elevation interval might be located in different areas of Switzerland. It might also be explained by positive RCM precipitation and negative RCM temperature biases at high elevations of the Alps (e.g., Kotlarski et al., 2015). At lower elevations, the station network is more balanced and the observations are probably more representative of the respective elevation interval. Despite a clear positive snowfall bias in mid-winter, the RCMs are generally able to reproduce the mean seasonal cycle of snowfall for elevations between 950 m a.s.l. - 1650 m a.s.l. (Fig. 3b). The fact that the major patterns of both the snowfall-elevation relationship and the mean seasonal snowfall cycle are basically represented indicates the general and physically consistent applicability of RCM output to assess future changes in mean and heavy Alpine snowfall. However, substantial biases in snowfall amounts are apparent and a bias correction of simulated snowfall seems to be required prior to the analysis of climate change signals of individual snowfall indices.

### 3.2 Calibration of bias correction

The analysis of total precipitation ratios (RCM simulations with respect to observations) for the EVAL period, which are computed to carry out the first step of the bias correction procedure, reveals substantial elevation dependencies. All simulations tend to overestimate total precipitation at high elevations (Fig. S1). This fact might ultimately be connected to an overestimation of surface snow amount in several EURO-CORDEX RCMs as reported by Terzago et al. (2017). As the precipitation ratio between simulations and observations approximately linearly depends on elevation, the calculation of $P_{AF}$ via a linear regression of the ratios against elevation (see Sec. 2.6) seems reasonable. By taking the inverse of this linear relation, $P_{AF}$ for every model and elevation can be derived. For the CCLM and RACMO simulations, these correction factors do not vary much with height, while $P_{AF}$ for MPI-ESM - REMO and EC-EARTH - HIRHAM is much larger than 1 in low lying areas, indicating a substantial underestimation of observed precipitation sums (Fig. 4a). However, for most elevations and simulations, $P_{AF}$ is generally smaller than 1, i.e., total precipitation is overestimated by the models. Similar model biases in the winter and spring seasons have already



been reported in previous works (e.g., Rajczak et al., in prep.; Smiatek et al., 2016). Especially at high elevations, these apparent positive precipitation biases could be related to observational undercatch, i.e., an underestimation of true precipitation sums by the observational analysis. Frei et al. (2003) estimated seasonal Alpine precipitation undercatch for three elevation intervals. Results show that measurement biases are largest in winter and increase with altitude. However, a potential undercatch can only partly explain the overestimation of precipitation found in the present work.

After applying $P_{AF}$ to the daily precipitation fields, a snowfall fractionation at the initial $T^*$ of 2 °C (see Eq. (4)) would lead to a snowfall excess in all 14 simulations as models typically experience a cold winter temperature bias. To match the observation-based and spatio-temporally averaged reference snowfall below 2750 m a.s.l., $T^*$ for all models needs to be decreased during the second step of the bias correction (Fig 4b). The adjusted $T^*_a$ values indicate a clear positive relation with the mean temperature bias in the EVAL period. This feature is expected since the stronger a particular model's cold bias the stronger the required adjustment of the snow fractionation temperature $T^*$ towards lower values in order to avoid a positive snowfall bias. Note that precipitation and temperature biases heavily depend on the GCM-RCM chain and seem to be rather independent from each other. While EC-EARTH – RACMO, for instance, shows one of the best performances in terms of total precipitation, its temperature bias of close to -5 °C is the largest deviation in our set of simulations. Concerning the partly substantial temperature biases of the EURO-CORDEX models shown in Figure 4 b, their magnitude largely agrees with Kotlarski et al. (2014; in reanalysis-driven simulations) and Smiatek et al. (2016).

**3.3 Evaluation of snowfall indices**

We next assess the performance of the bias correction procedure by comparing snowfall indices derived from separated and bias-corrected RCM snowfall amounts against the observation-based reference. The period for which this comparison is carried out is EVAL, i.e., it is identical to the calibration period of the bias correction. We hence do not intend a classical cross validation exercise with separate calibration and validation periods, but try to answer the following two questions: (a) Which aspects of the Alpine snowfall climate are corrected for, and (b) for which aspects do biases remain even after application of the bias correction procedure.

Figure 5 shows the evaluation results of the six snowfall indices based on the separated and not bias-corrected simulated snowfall ($RCM_{sep+nbc}$), and the separated and bias-corrected simulated snowfall ($RCM_{sep+bc}$). In the first case the snowfall separation of raw precipitation is performed with $T^*=2°C$, while in the second case precipitation is corrected and the separation is performed with a bias-adjusted temperature $T^*_a$. The first column represents the mean September to May statistics, while columns 2-4 depict the seasonal cycle at monthly resolution for three distinct elevation intervals.

The analysis of $S_{mean}$ confirms that $RCM_{sep+bc}$ is able to reproduce the observation-based reference in the domain mean as well as in most individual elevation intervals. The domain-mean agreement is a direct consequence of the design of the bias correction procedure (see above). $RCM_{sep+nbc}$, on the other hand, consistently overestimates $S_{mean}$ by up to a factor of 2.5 as a consequence of positive





precipitation and negative temperature biases (cf. Fig. 4). Also the seasonal cycle of $S_{mean}$ for
$RCM_{sep+bc}$ yields a satisfying performance across all three elevation intervals, while $RCM_{sep+nbc}$ tends
to produce too much snowfall over all months and reveals an increasing model spread with elevation.
For the full domain and elevations around 1000 m, the observation-based reference indicates a mean
$S_{freq}$ of 20% between September and May. Up to 1000 m a.s.l. $RCM_{sep+bc}$ reflects the increase of this
index with elevation adequately. However, towards higher elevations the approximately constant $S_{freq}$
of 30% in the reference is not captured by the simulation-derived snowfall. Notably during wintertime,
both $RCM_{sep+bc}$ and $RCM_{sep+nbc}$ produce too many snowfall days, i.e., overestimate snowfall
frequency. This feature is related to the fact that climate models typically tend to overestimate the wet
day frequency over the Alps especially in wintertime (Rajczak et al., 2013) and that the bias correction
procedure employed does not explicitly correct for potential biases in precipitation frequency. Due to
the link between mean snowfall on one side and snowfall frequency and mean intensity on the other
side, opposite results are obtained for the mean snowfall intensity $S_{int}$. $RCM_{sep+bc}$ largely
underestimates mean intensities during snowfall days while $RCM_{sep+nbc}$ typically better reflects the
reference. Nevertheless, deviations during winter months at mid-elevations are not negligible. Mean
September-May $S_{frac}$ in the reference exponentially increases with elevation. This behaviour is
reproduced by both $RCM_{sep+bc}$ and $RCM_{sep+nbc}$. Notwithstanding, $RCM_{sep+bc}$ results are more accurate
compared to $RCM_{sep+nbc}$, which turns out to be biased towards too large snowfall fractions.
For the two heavy snowfall indices $S_{q99}$ and $S_{1d}$, $RCM_{sep+nbc}$ appears to typically match the reference
better than $RCM_{sep+bc}$. Especially at high elevations, $RCM_{sep+bc}$ produces too low snowfall amounts.
This again highlights the fact that the bias correction procedure is designed to correct for biases in
mean snowfall, but does not necessarily improve further aspects of the simulated snowfall climate.
The spatial patterns of $S_{mean}$ for the 14 $RCM_{sep+bc}$ simulations from September to May are presented in
Figure 6. The observational-based reference (lower right panel) reveals a snowfall distribution with
highest values along the Alpine main ridge, whereas the Swiss plateau, Southern Ticino and main
valleys such as the Rhône and Rhine valley experience less snowfall. Almost all bias-corrected
models are able to represent the overall picture with snow-poor lowlands and snow-rich Alpine
regions. Nevertheless substantial differences to the observations concerning the spatial snowfall
pattern can arise. EC-EARTH - HIRHAM, for example, is subject to a "pixelated" structure. This could
be the result of frequent grid-cell storms connected to parameterisations struggling with complex
topographies. Such inaccuracies in the spatial pattern are not corrected for by our simple bias
correction approach that only targets domain-mean snowfall amounts at elevations below 2750 m
a.s.l. and that does not considerably modify the simulated spatial snowfall patterns.. Note that these
patterns are obviously strongly determined by the RCM itself and only slightly depend on the driving
GCM (see, for instance, the good agreement among the CCLM and the RCA simulations).
In summary, after applying the bias correction to the simulations most snowfall indices are fairly well
represented at elevations below 1000 m a.s.l.. With increasing altitude and smaller sample sizes in
terms of number of grid cells, reference and $RCM_{sep+bc}$ diverge. This might be caused by the remaining
simulated overestimation of $S_{freq}$ and an underestimation of $S_{int}$. While the bias correction approach





leads to a reduction of $S_{int}$ due to the total precipitation adjustment, $S_{freq}$ is only slightly modified by this
correction and by the adjustment of $T^*$. Nevertheless, these two parameters strongly influence other
snowfall indices. The counteracting effects of overestimated $S_{freq}$ and underestimated $S_{int}$ result in
appropriate amounts of $S_{mean}$ whereas discrepancies for $S_{q99}$ and $S_{1d}$ are mainly driven by the
underestimation of $S_{int}$.
**4 Snowfall projections for the late 21$^{st}$ century**
For the study of climate change signals, the analysis domain is extended to the entire Alps (see Sec.
2.3). Due to the identified difficulties of bias correcting certain snowfall indices (see Sec 3.3), emphasis
is laid upon relative signals of change (see Eq. 2). This type of change can be expected to be less
dependent on the remaining inaccuracies after the correction. If not stated otherwise, all results in this
Section are based on the RCM$_{sep+bc}$ data, i.e., on separated and bias-corrected RCM snowfall, and on
the RCP8.5 emission scenario.
Projections for seasonal $S_{mean}$ show a considerable decrease over the entire Alpine domain (Fig. 7).
Most RCMs project largest percentage losses of more than 80% across the Alpine forelands such as
the Po Valley or Western France. Over the Alpine ridge, reductions are smaller but still mostly
negative. Elevated regions between Southeastern Switzerland, Northern Italy and Austria seem to be
least affected by the overall snowfall reduction. Some of the simulations (e.g., CNRM-RCA, MPI-
ESM-RCA or MPI-ESM-REMO) project only minor changes in these regions. Experiments employing
the same RCM but different driving GCMs (e.g. the four simulations of RCA), but also experiments
employing the same GCM but different RCMs (e.g. the four simulations driven by EC-EARTH) can
significantly disagree in regional-scale change patterns and especially in the general magnitude of
change.  This highlights a strong influence of both the driving GCMs and the RCMs themselves on
snowfall changes, representing effects of  large-scale circulation and meso-scale response,
respectively.
A more detailed analysis is provided in Fig.8 that addresses the vertical and seasonal distribution of
snowfall changes. It reveals that relative (seasonal mean) changes of $S_{mean}$ appear to be strongly
dependent on elevation (Fig.8, top left panel). The multimodel mean change ranges from -80% at low
elevations to -10% above 3000 m a.s.l.. Largest differences between neighbouring elevation intervals
are obtained from 750 m a.s.l. to 1500 m a.s.l.. Over the entire Alps, the results show a reduction of
$S_{mean}$ by -35% to -55% with a multimodel mean of -45%. The multimodel spread appears to be rather
independent of elevation and is comparably small, confirming that, overall, the spatial distributions of
the change patterns are similar across all model chains (cf. Fig. 7). All simulations point to decreases
over the entire nine-month period September to May for the two elevation intervals <1000 m a.s.l. and
1000 to 2000 m a.s.l.. Above 2000 m a.s.l., individual simulations show an increase of $S_{mean}$ by up to
20% in mid-winter which forces the multimodel mean change to be slightly positive in January and
February.
Decreases of $S_{freq}$ are very similar to change sin mean snowfall. Mean September-May changes are
largest below 1000 m a.s.l., while differences among elevation intervals become smaller in the upper



part. In-between is a transition zone with rather strong changes with elevation. Individual simulations
with large reductions in $S_{mean}$, such as the RCA experiments, also project strongest declines in $S_{freq}$. In
contrast, the mean snowfall intensity $S_{int}$ is subject to smallest percentage variations in our set of
snowfall indices. Strong percentage changes for some models in September are due to the small
sample size (only few grid points considered) and the low snowfall amounts in this month. Apart from
mid elevations with decreases of roughly -10%, mean intensities from September to May are projected
to remain almost unchanged by the end of the century. For both seasonal and monthly changes,
model agreement is best for high elevations while the multimodel spread is largest for lowlands. Large
model spread at low elevations might be caused by the small number of grid points used for averaging
over the respective elevation interval, especially in autumn and spring.
Similar results are obtained for the heavy snowfall indices $S_{q99}$ and $S_{1d}$. While percentage decreases
at lowermost elevations are even larger than for $S_{mean}$, losses at high elevations are less pronounced,
resulting in similar domain-mean change signals for heavy and mean snowfall. Substantial differences
between monthly $\delta S_{q99}$ and $\delta S_{1d}$ appear at elevations below 1000 m a.s.l.. Here, percentage losses of
$S_{q99}$ are typically slightly more pronounced. Above 2000 m a.s.l. both indices appear to remain almost
constant between January and March with change signals close to zero. The multimodel mean
changes even hint to slight increases of both indices. Concerning changes in the snowfall fraction, i.e.,
in the relative contribution of snowfall to total precipitation, our results indicate that current seasonal
and domain mean $S_{frac}$ might drop by about -50% (Fig. 8, lowermost row). Below 1000 m a.s.l., the
strength of the signal is almost independent of the month, and mutlimodel average changes of the
snow fraction of about -80% are obtained. At higher elevations changes during mid-winter are less
pronounced compared to autumn and spring but still negative.

## 5 Discussion

### 5.1 Effect of temperature, snowfall frequency and intensity on snowfall changes

The results in Section 4 indicate substantial changes of snowfall indices over the Alps in regional
climate projections. With complementary analyses presented in Figures 9 and 10 we shed more light
on the responsible mechanisms, especially concerning projected changes in mean and heavy
snowfall. For this purpose Figures 9a-b,e-f show the relationship of both mean and heavy snowfall
amounts in the CTRL period and their respective percentage changes with the climatological CTRL
temperature of the respective (climatological) month, elevation interval and GCM-RCM chain. For
absolute amounts ($S_{mean}$, $S_{q99}$; Fig. 9a,e) a clear negative relation is found, i.e., the higher the CTRL
temperature the lower the snowfall amounts. For $S_{mean}$ the relation levels off at mean temperatures
higher than about 6°C with mean snowfall amounts close to zero. For temperatures below about -6°C
a considerable spread in snowfall amounts is obtained, i.e., mean temperature does not seem to be
the controlling factor here. Relative changes of both quantities (Fig. 9b,f), however, are strongly
controlled by the CTRL period's temperature level with losses close to 100% for warm climatic settings
and partly increasing snowfall amounts for colder climates. This dependency of relative snowfall
changes on CTRL temperature is in line with previous works addressing future snowfall changes on



both hemispheric and regional scales (de Vries et al., 2014; Krasting et al., 2013; Räisänen, 2016). The spread of changes within a given CTRL temperature bin can presumably be explained by the respective warming magnitudes that differ between elevations, months and GCM-RCM chains. About half of this spread can be attributed to the month and the elevation alone (compare the spread of the black markers to the one of the red markers which indicate multimodel averages).

For most months and elevation intervals, percentage reductions in $S_{mean}$ and $S_{q99}$ reveal an almost linear relationship with $\delta S_{freq}$ (Fig. 9c, g). The decrease of $S_{freq}$ with future warming can be explained by a shift of the temperature probability distribution towards higher temperatures, leading to fewer days below the freezing level (Fig. 10, top row). Across the three elevation intervals <1000 m a.s.l., 1000-2000 m a.s.l. and > 2000, relative changes in the number of days with temperatures below the freezing level (T≤0°C) are in the order of -65%, -40% and -20%, respectively (not shown). This approximately corresponds to the simulated decrease of $S_{freq}$ (cf. Fig 8), which in turn, is of a similar magnitude as found in previous works addressing future snowfall changes in the Alps (Schmucki et al., 2015b; Zubler et al., 2014). Due to the general shift of the temperature distribution and the "loss" of very cold days (Fig. 10, top row) future snowfall furthermore occurs in a narrower temperature range (Fig. 10, second row).

Contrasting this general pattern of frequency-driven decreases of both mean and heavy snowfall, no changes or even slight increases of $S_{mean}$, $S_{q99}$ and $S_{1d}$ at high elevations are expected in mid-winter (see Fig. 8). This can to some part be explained by the general increase of total winter precipitation (Rajczak et al., in prep; Smiatek et al., 2016) that obviously offsets the warming effect in high-elevation regions where a substantial fraction of the future temperature PDF is still located below the rain-snow transition (Fig. 10, top row). This process has also been identified in previous works to be, at last partly, responsible for future snowfall increases (de Vries et al., 2014; Krasting et al., 2013; Räisänen, 2016). Furthermore, the magnitude of the increases of both mean and heavy snowfall is obviously driven by positive changes of $S_{int}$, while $S_{freq}$ remains constant (Fig. 9c,g). An almost linear relationship between positive changes of $S_{int}$ and positive changes of $S_{mean}$ and $S_{q99}$ is obtained (Fig. 9d,h; upper right quadrants. Nevertheless, the high-elevation mid-winter growth in $S_{mean}$ is smaller than the identified increases of mean winter total precipitation. This can be explained by the persistent decrease of $S_{frac}$ during the cold season (see Fig. 8, lowermost row).

For elevation intervals with simulated monthly temperatures between -6°C and 0°C in the CTRL period, $S_{mean}$ appears to decrease stronger than $S_{q99}$ (cf. Fig. 9b,f). O'Gorman (2014) found a very similar behaviour when analysing mean and extreme snowfall projections over the Northern Hemisphere within a set of GCMs. This finding is related to the fact that future snowfall decreases are mainly governed by a decrease of snowfall frequency while snowfall increases in high-elevated regions in mid-winter seem to be caused by increases of snowfall intensity. It can obviously be explained by the insensitivity of the temperature interval at which extreme snowfall occurs to climate warming and by the shape of the temperature – snowfall intensity distribution itself (Fig. 10, third row). The likely reason behind positive changes of $S_{int}$ at high-elevated and cold regions is the higher water holding capacity of the atmosphere in a warmer climate. According to the Clausius-Clapeyron relation,



saturation vapour pressure increases by about 7% per degree warming (Held and Soden, 2006). Previous studies have shown that simulated changes of heavy and extreme precipitation are consistent with this theory (e.g., Allen and Ingram, 2002; Ban et al., 2015). In terms of snowfall, we find the Clausius-Clapeyron relation to be applicable for negative temperatures up to approximately -5°C as well (Fig. 10, third row, dashed lines). Inconsistencies for temperatures between -5°C and 0°C are due to a snow fraction $sf$ < 100% for corresponding precipitation events.

For further clarification, Figure 11 schematically illustrates the governing processes behind the changes of mean and heavy snowfall that differ between climatologically warm (decreasing snowfall) and climatologically cold climates (increasing snowfall). As shown in Figure 10 (third row), the mean $S_{int}$ distribution is rather independent on future warming and similar temperatures are associated with similar mean snowfall intensities. In particular, heaviest snowfall is expected to occur slightly below the freezing level in both the CTRL and the SCEN period (Fig. 11a). How often do such conditions prevail in the two periods? In a warm current climate, i.e., at low elevations or in the transition seasons, heavy snowfall only rarely occurs as the temperature interval for highest snowfall intensity is already situated in the left tail of the CTRL period's temperature distribution (Fig. 11b). With future warming, i.e., with a shift of the temperature distribution to the right, the probability for days to occur in the heavy snowfall temperature interval (dark grey shading) decreases stronger than the probability of days to occur in the overall snowfall regime (light gray shading). This results in (1) a general decrease of snowfall frequency, (2) a general decrease of mean snowfall intensity and (3) a general and similar decrease of both mean and heavy snowfall amounts. In contrast, at cold and high-elevated sites CTRL period temperatures are often too low to trigger heavy snowfall since a substantial fraction of the temperature PDF is located to the left of the heavy snowfall temperature interval (Fig. 11 c). The shifted distribution in a warmer SCEN climate, however, peaks within the temperature interval that favours heavy snowfall. This leads to a probability increase for days to occur in the heavy snowfall temperature range despite the general reduction in $S_{freq}$ (lower overall probability of days to occur in the entire snowfall regime, light gray). As a consequence, mean $S_{int}$ tends to increase and the reduction of heavy snowfall amounts is less pronounced (or even of opposing sign) than the reduction in mean snowfall. For individual (climatologically cold) regions and seasons, the increase of mean $S_{int}$ might even compensate the $S_{freq}$ decrease, resulting in an increase of both mean and heavy snowfall amounts. Note that in a strict sense these explanations only hold in the case that the probability of snowfall to occur at a given temperature does not change considerably between the CTRL and the SCEN period. This is approximately given (Fig. 10, bottom row), which presumably indicates only minor contributions of large scale circulation changes and associated humidity changes on both the temperature - snowfall frequency and the temperature - snowfall intensity relation.

## 5.2 Emission scenario uncertainty

The projections presented in the previous sections are based on RCP8.5, but depend on the emission scenario considered. To assess this type of uncertainty we here compare the $RCM_{sep+bc}$ simulations for the previously shown RCP8.5 emission scenario against those assuming the more moderate RCP4.5 scenario. As a general picture, the weaker RCP4.5 scenario is associated with less pronounced changes of snowfall indices (Fig. 12). Differences in mean seasonal $\delta S_{mean}$ between the two emission





scenarios are most pronounced below 1000 m a.s.l. where percentage changes for RCP4.5 are about one third smaller than for RCP8.5. At higher elevations, multimodel mean changes better agree and the multimodel ranges for the two emission scenarios start overlapping, i.e., individual RCP4.5 experiments can be located in the RCP8.5 multimodel range and vice versa. Over the entire Alpine domain, about -25% of current snowfall is expected to be lost under the moderate RCP4.5 emission scenario while a reduction of approximately -45% is projected for RCP8.5. For seasonal cycles, the difference of $\delta S_{mean}$ between RCP4.5 and RCP8.5 is similar for most months and slightly decreases with altitude. Above 2000 m a.s.l., the simulated increase of $S_{mean}$ appears to be independent of the chosen RCP in January and February, while negative changes before and after mid-winter are more pronounced for RCP8.5. Alpine domain mean $\delta S_{q99}$ almost doubles under the assumption of stronger GHG emissions. This is mainly due to differences at low elevations whereas above 2000 m a.s.l. $\delta S_{q99}$ does not seem to be strongly affected by the choice of the emission scenario. Differences in monthly mean changes are in close analogy to $\delta S_{mean}$. Higher emissions lead to a further negative shift in $\delta S_{q99}$. Up to mid-elevations differences are rather independent of the season. However, at highest elevations and from January to March, differences between RCP4.5 and RCP8.5 are very small.

Despite the close agreement of mid-winter snowfall increases at high elevations between the two emission scenarios, obvious differences in the spatial extent of the region of mean seasonal snowfall increases can be found (cf Figs. S5 and 7 for $\delta S_{mean}$, and Figs. S6 and S7 for $\delta S_{q99}$). In most simulations, the number of grid cells along the main Alpine ridge that show either little change or even increases of seasonal mean $S_{mean}$ or $S_{q99}$ is larger for RCP4.5 than for RCP8.5 with its larger warming magnitude.

### 5.3 Intercomparison of projections with separated and raw snowfall

An intercomparison of relative change signals for $RCM_{sep+bc}$ (separated and bias-corrected), $RCM_{sep+nbc}$ (separated and not bias-corrected) and simulated raw snowfall output ($RCM_{raw}$) based on the nine RCMs providing raw snowfall as output variable (see Tab. 1) reveals no substantial differences (Fig. 13, top row). In the three data sets, multimodel mean relative changes are very similar for all analysed snowfall indices and elevation intervals. Furthermore, multimodel mean differences between $RCM_{sep+bc}$, $RCM_{sep+nbc}$ and $RCM_{raw}$ simulations are smaller than the corresponding multimodel spread of $RCM_{sep+bc}$ simulations and emission scenario uncertainties (cf. Figs. 12, 13 and S8).

This finding is in contrast to absolute change characteristics (Fig. 13, bottom row). Results based on the three data sets agree in the sign of change, but not in their magnitude, especially at high elevations >2000 m. As the relative changes are almost identical, the absolute changes strongly depend upon the treatment of biases in the control climate. In summary, these findings indicate that (a) the snowfall separation method developed in the present work yields rather good proxies for relative changes of snowfall indices in raw RCM output (which is for many GCM-RCM chains not available), and that (b) the additional bias-correction of separated snowfall amounts only has a weak influence on relative change signals of snowfall indices, but can have substantial effects on absolute changes.





## 6 Conclusions and outlook

The present work makes use of state-of-the-art EURO-CORDEX RCM simulations to assess changes of snowfall indices over the European Alps by the end of the 21st century. For this purpose, snowfall is separated from total precipitation using near-surface air temperature in both the RCMs and in the observations on a daily basis. The analysis yields a number of robust signals, consistent across a range of climate model chains and across emission scenarios. Relating to the main objectives we find the following:

**Snowfall separation on an RCM grid.** Binary snow fractionation with a fixed temperature threshold on coarse-resolution grids (with 11 km resolution) leads to an underestimation of mean snowfall and an overestimation of heavy snowfall. To overcome these deficiencies, the Richards snow fractionation method is implemented. This approach expresses that the coarse-grid snow fraction depends not only on daily mean temperature, but also on topographical subgrid-scale variations. Accounting for the latter results in better estimates for mean and heavy snowfall. However, due to limited observational coverage the parameters of this method are fitted for Switzerland only and are then applied to the entire Alpine domain. Whether this spatial transfer is robust could further be investigated by using observational data sets covering the full domain of interest but is out of the scope of this study.

**Snowfall bias correction.** Simulations of the current EURO-CORDEX ensemble are subject to considerable biases in precipitation and temperature, which translate into biased snowfall amounts. In the EVAL period, simulated precipitation is largely overestimated, with increasing biases toward higher altitudes. On the other hand, simulated near surface temperatures are generally too low with largest deviations over mountainous regions. These findings were already reported in previous studies (e.g. Frei et al., 2003; Kotlarski et al., 2012; Kotlarski et al., 2015; Rajczak et al., 2013; Smiatek et al., 2016). By implementing a simple bias correction approach, we are able to partly reduce these biases and the associated model spread, which should enable more robust change estimates. The corrected model results reproduce the seasonal cycles of mean snowfall fairly well. However, substantial biases remain in terms of heavy snowfall, snowfall intensities (which in general are overestimated), snowfall frequencies, and spatial snowfall distributions. Further improvements might be feasible by using more sophisticated bias correction methods, such as quantile mapping (e.g., Rajczak et al., 2016), local intensity scaling of precipitation (e.g., Schmidli et al., 2006), or weather generators (e.g. Keller at al., 2016). Advantages of the approach employed here are its simplicity, its direct linkage to the snowfall separation method and, as a consequence, its potential ability to account for non-stationary snowfall biases. Furthermore, a comparison to simulated raw snowfall for a subset of nine simulations revealed that relative change signals are almost independent of the chosen post-processing strategy.

**Snowfall projections for the late 21st century.** Snowfall climate change signals are assessed by deriving the changes in snowfall indices between the CTRL period 1981 - 2010 and the SCEN period 2070 - 2099. Our results show that by the end of the 21st century, snowfall over the Alps will be considerably reduced. Between September and May mean snowfall is expected to decrease by approximately -45% (multimodel mean) under an RCP8.5 emission scenario. For the more moderate RCP4.5 scenario, multimodel mean projections show a decline of -25%. These results are in good



agreement with previous works (e.g. de Vries et al., 2014; Piazza et al., 2014, Räisänen, 2016). Low-
lying areas experience the largest percentage changes of more than -80%, while the highest Alpine
regions are only weakly affected. Variations of heavy snowfall, defined by the 99% all-day snowfall
percentile, show at low-lying elevations an even more pronounced signal. With increasing elevation,
percentage changes of heavy snowfall are generally smaller than for mean snowfall. O'Gorman (2014)
found a very similar behaviour by analysing projected changes in mean and extreme snowfall over the
entire Northern Hemisphere. He pointed out that heavy and extreme snowfall occurs near an optimal
temperature (near or below freezing, but not too cold), which seems to be independent of climate
warming. We here confirm this conclusion. At mid and high elevations the optimal temperature for
heavy snowfall will still occur in a warmer climate and, hence, heavy snowfall amounts will decrease
less strongly compared to mean snowfall, and may even increase in some areas..
At first approximation, the magnitude of future warming strongly influences the reduction of mean and
heavy snowfall by modifying the snowfall frequency. Snowfall increases may however occur at high
(and thus cold) elevations, and these are not caused by frequency changes. Here, snowfall increases
due to (a) a general increase of total winter precipitation combined with only minor changes in snowfall
frequency, and (b) more intense snowfall. This effect has a pronounced altitudinal distribution and may
be particularly strong under conditions (depending upon location and season) where the current
climate is well below freezing. Such conditions may experience a shift towards a more snowfall-
friendly temperature range (near or below freezing, but not too cold) with corresponding increases of
mean snowfall, despite a general decrease of the snowfall fraction.
The identified future changes of snowfall over the Alps can lead to a variety of impacts in different
sectors. With decreasing snowfall frequencies and the general increase of the snowline (e.g.,
Beniston, 2003; Gobiet et al., 2014; Hantel et al., 2012), both associated with temperature changes,
ski lift operators are looking into an uncertain future. A shorter snowfall season will likely put them
under greater financial pressure. Climate change effects might be manageable only for ski areas
reaching up to high elevations (e.g. Elsasser and Bürki, 2002). Even so these resorts might start later
into the ski season, the snow conditions into early spring could change less dramatically due to
projected high-elevation snowfall increases in mid-winter. A positive aspect of the projected decrease
in snowfall frequency might be a reduced expenditures for airport and road safety (e.g., Zubler et al.,
711  2015).

At lower altitudes, an intensification of winter precipitation, combined with smaller snowfall fractions
(Serquet et al., 2013), increases the flood potential (Beniston, 2012). Snow can act as a buffer by
releasing melt water constantly over a longer period of time. With climate warming, this storage
capacity is lost, and heavy precipitation immediately drains into streams and rivers which might not be
able to take up the vast amount of water fast enough. Less snowmelt will also have impacts on
hydropower generation and water management (e.g., Weingartner et al., 2013). So far, many Alpine
regions are able to bypass dry periods by tapping melt water from mountainous regions. With reduced
snow-packs due to less snowfall, water shortage might become a serious problem in some areas.



Regarding specific socio-economic impacts caused by extreme snowfall events, conclusions based on the results presented in this study are difficult to draw. It might be possible that the 99% all-day snowfall percentile we used for defining heavy snowfalls, is not appropriate to speculate about future evolutions of (very) rare events (Schär et al., 2016). To do so, one might consider applying a generalized extreme value (GEV) analysis which is more suitable for answering questions related to rare extreme events.

## 7 Data Availability

The EURO-CORDEX RCM data analysed in the present work are publicly available - parts of them for non-commercial use only - via the Earth System Grid Federation archive (ESGF; e.g., https://esgf-data.dkrz.de). The observational datasets RHiresD and TabsD as well as the snow depth data for Switzerland are available for research and educational purposes from kundendienst@meteoschweiz.ch. The analysis code is available from the corresponding author on request.

## 8 Competing Interests

The authors declare that they have no conflict of interest.

## 9 Acknowledgements

We gratefully acknowledge the support of Jan Rajczak, Urs Beyerle and Curdin Spirig (ETH Zurich) as well as Elias Zubler (MeteoSwiss) in data acquisition and pre-processing. Christoph Frei (MeteoSwiss) and Christoph Marty (WSL-SLF) provided important input on specific aspects of the analysis. Finally, we thank the climate modelling groups of the EURO-CORDEX initiative for producing and making available their model output.

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





**Figures**





**Figure 1** Topography at EUR-11 (0.11°) RCM resolution of the Alpine domain used for the assessment of
snowfall projections. The bold black outline marks the Swiss sub-domain used for the assessment of the bias
correction approach.





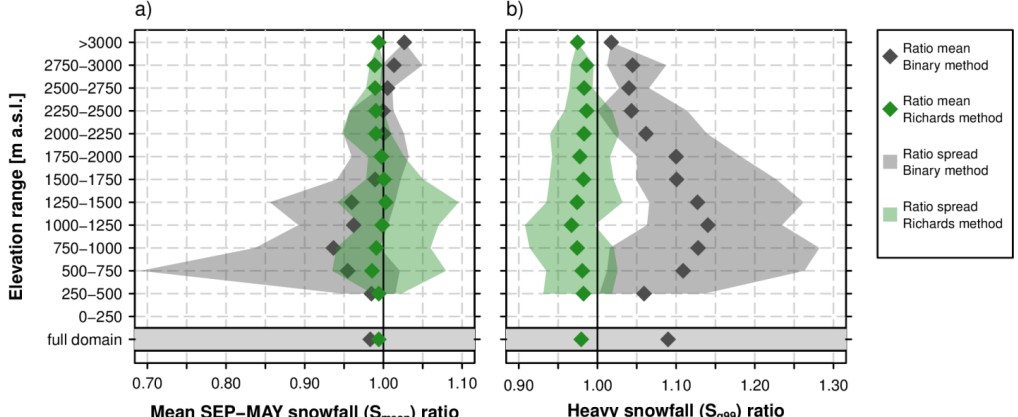



**Figure 2** Snowfall ratios for the Binary and Richards snow factionation method (ratio between the snowfall of the
respective method and the full subgrid snow representation). The ratios are valid at the course-resolution grid (12
km). a) Ratios for mean snowfall, $S_{mean}$. b) Ratios for heavy snowfall, $S_{q99}$. Ratio means were derived after
averaging the corresponding snowfall index for 250 m elevation intervals in Switzerland while the ratio spread
represents the minimum and maximum grid point-based ratios in the corresponding elevation interval. This
analysis is entirely based on the observational data sets TabsD and RhiresD.





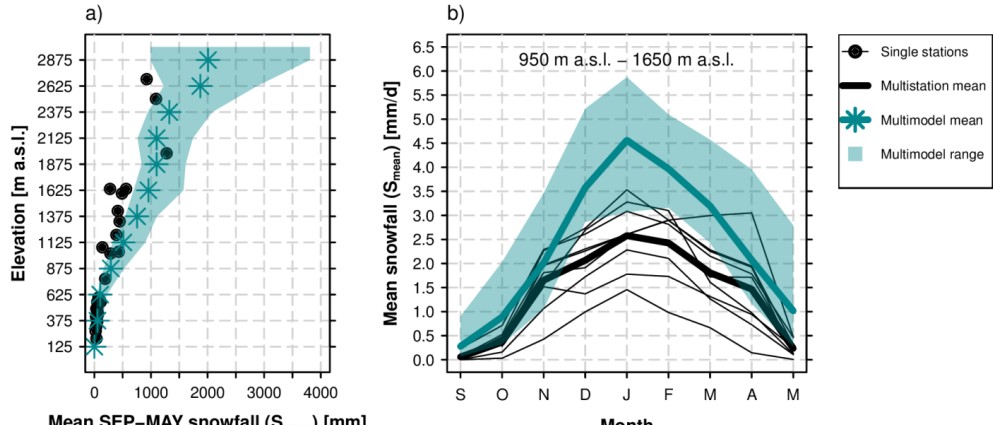

**Figure 3** Comparison of measured fresh snow sums of 29 MeteoSwiss stations vs. simulated RCM raw snowfall in Switzerland in the EVAL period 1971-2005. a) Mean September – May snowfall vs. elevation. The simulation data are based on the spatio-temporal mean of 250 m elevation ranges and plotted at the mean elevation of the corresponding interval. b) Seasonal September-May snowfall cycle for the elevation interval 950 m a.s.l. to 1650 m a.s.l.. Simulated multimodel means and spreads are based on a subset of 9 EURO-CORDEX simulations providing raw snowfall as output variable (see Tab. 1).





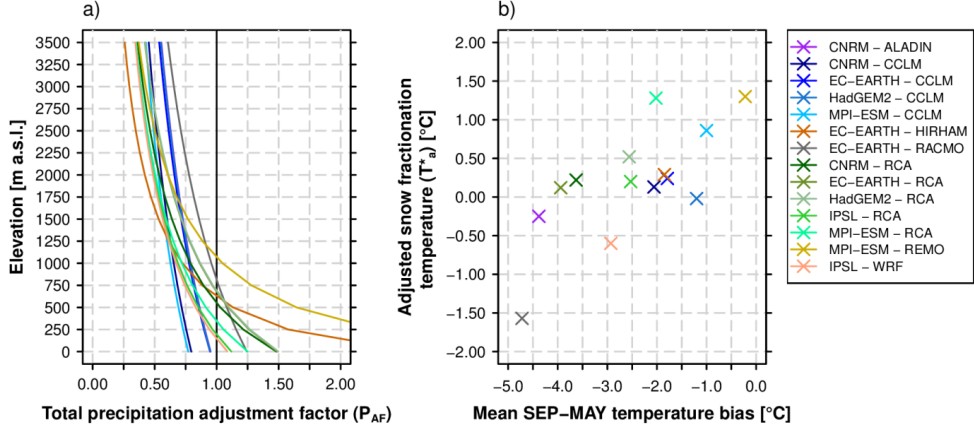

929

930

**Figure 4** Bias correction and adjustment factors. a) Elevation-dependent total precipitation adjustment factors,
$P_{AF}$, for the 14 GCM-RCM chains (see Eq. 10). b) Scatterplot of mean September to May temperature biases
(RCM simulation minus observational analysis) vs. adjusted snow fractionation temperatures, $T^*_a$.





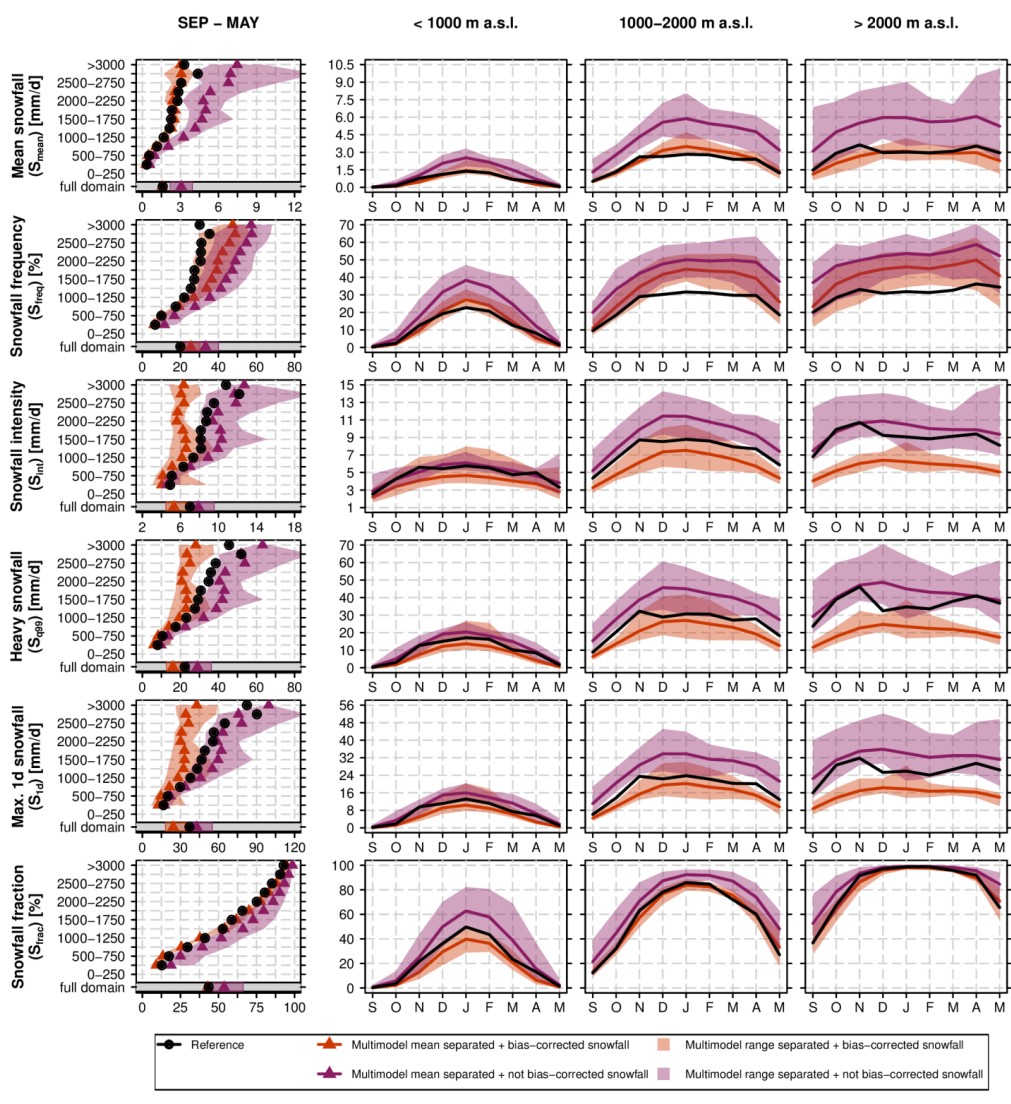

**Figure 5** Evaluation of snowfall indices in the EVAL period 1971-2005 for the 14 snowfall separated + bias corrected ($RCM_{sep+bc}$) and 14 snowfall separated + not bias corrected ($RCM_{sep+nbc}$) RCM simulations vs. observation-based reference. The first column shows the mean September-May snowfall index statistics vs. elevation while the monthly snowfall indices (spatially averaged over the elevation intervals <1000 m.a.s.l., 1000 m a.s.l.-2000 m a.s.l. and >2000 m a.s.l.) are displayed in columns 2-4.





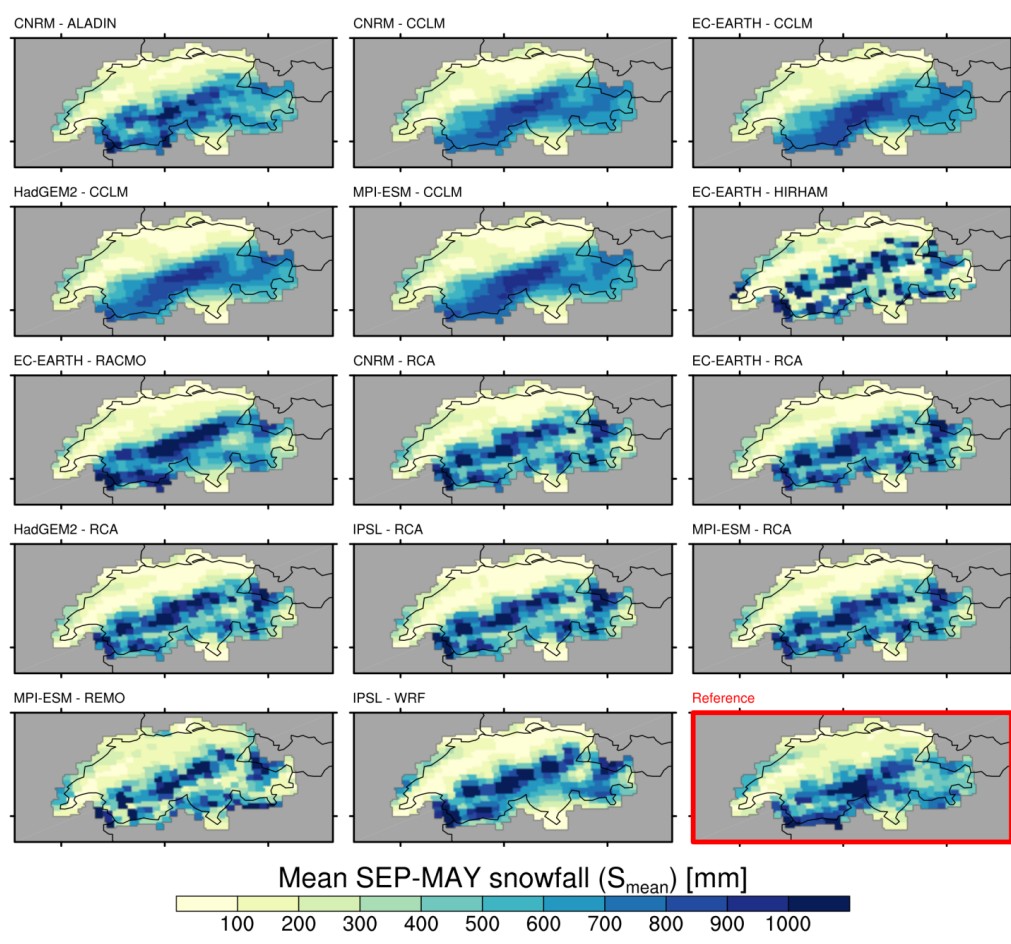

**Figure 6** Spatial distribution of mean September-May snowfall, $S_{mean}$, in the EVAL period 1971-2005 and for the 14 snowfall separated + bias corrected RCM simulations (RCM$_{sep+bc}$). In the lower right panel, the map of the observation-based reference is shown.





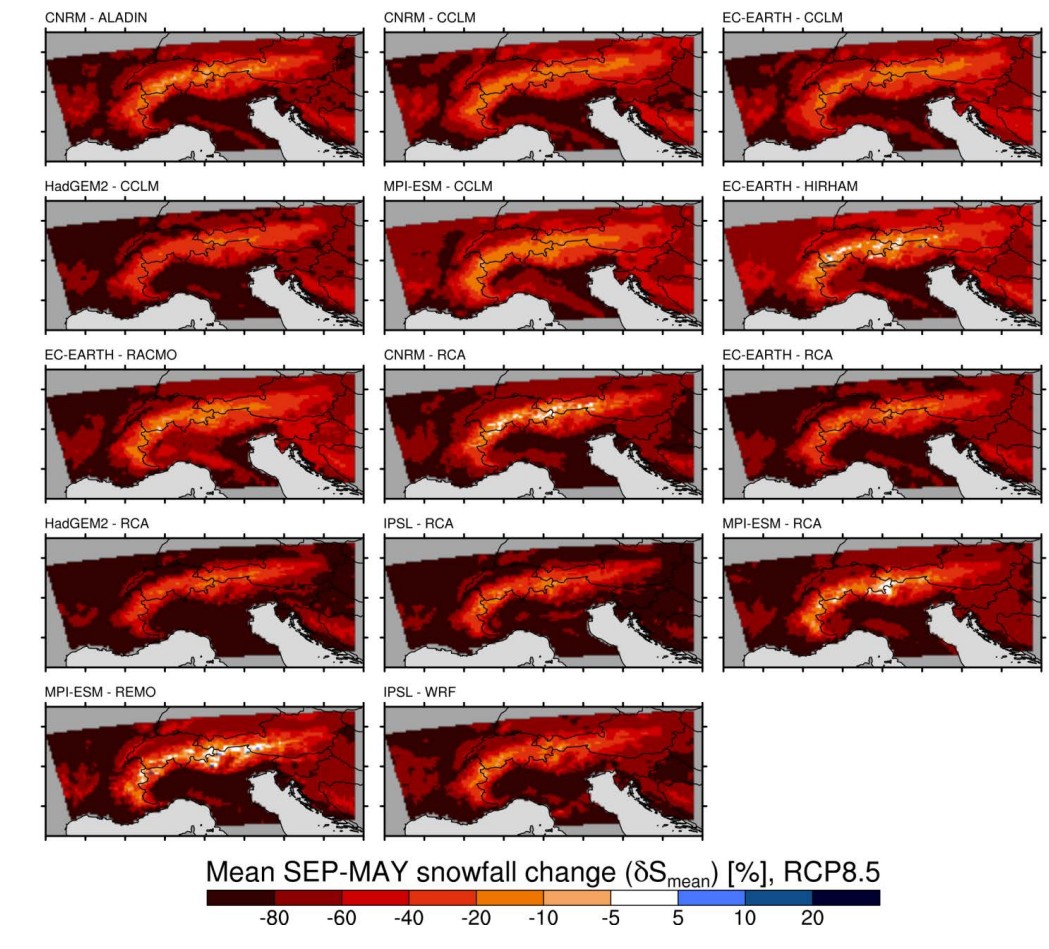



**Figure 7** Spatial distribution of relative changes (SCEN period 2070-2099 with respect to CTRL period 1981-
2010) in mean September-May snowfall, $\delta S_{mean}$, for RCP8.5 and for the 14 snowfall separated + bias corrected
RCM simulations (RCM$_{sep+bc}$). For RCP4.5, see Fig. S5.






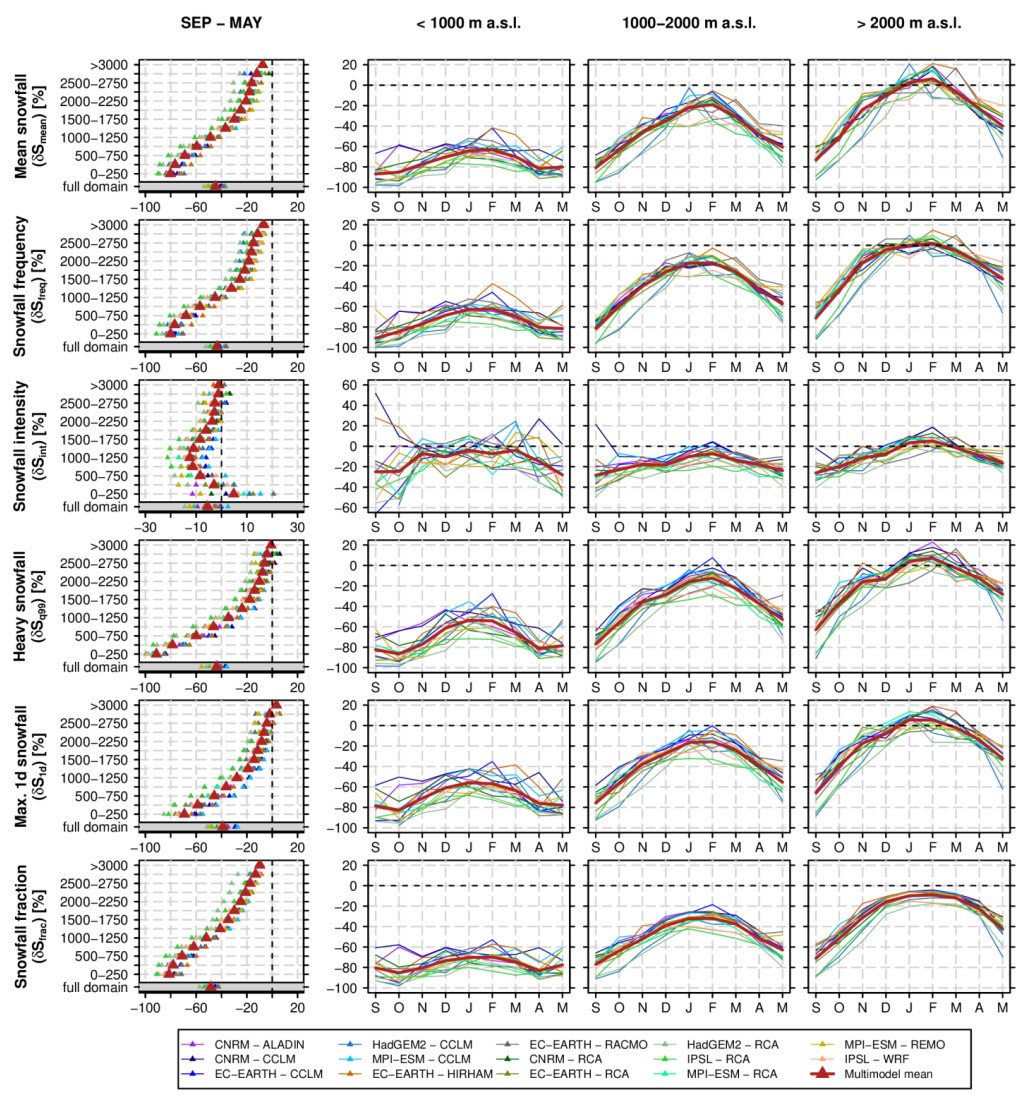



**Figure 8** Relative changes (SCEN period 2070-2099 with respect to CTRL period 1981-2010) of snowfall indices
based on the 14 snowfall separated + bias corrected RCM simulations (RCM$_{sep+bc}$) for RCP8.5. The first column
shows the mean September-May snowfall index statistics vs. elevation while monthly snowfall index changes
(spatially averaged over the elevation intervals <1000 m.a.s.l., 1000 m a.s.l.-2000 m a.s.l. and >2000 m a.s.l.) are
displayed in columns 2-4.

961





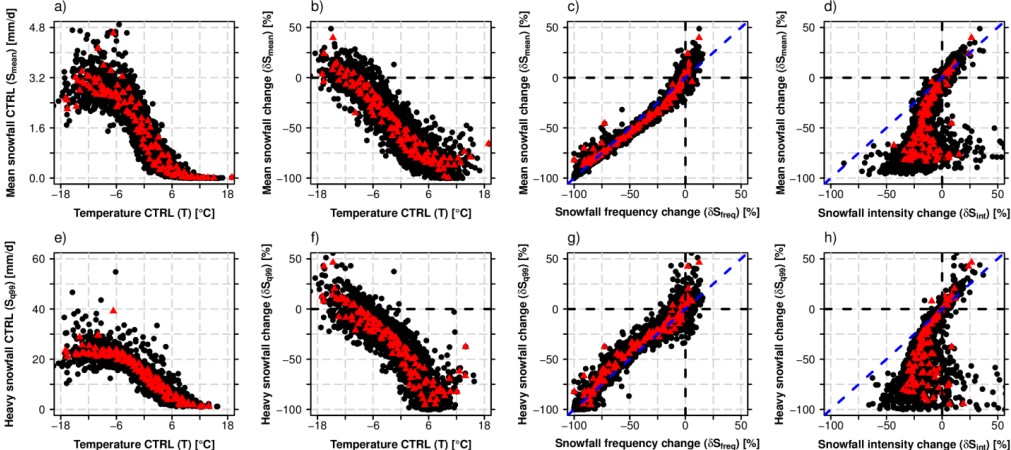

962

963

**Figure 9** Intercomparison of various snowfall indices and relationship with monthly mean temperature in CTRL.
For each panel, the monthly mean statistics for each 250 m elevation interval and for each of the 14 individual
GCM-RCM chains were derived (black circles). Red triangles denote the multimodel mean for a specific month
and elevation interval. The monthly statistics were calculated by considering all grid points of the specific
elevation intervals which are available for both variables in the corresponding scatterplot only (area consistency).
The data were taken from the 14 snowfall separated + bias corrected ($RCM_{sep+bc}$) RCM simulations. Relative
changes are based on the RCP8.5 driven simulations (SCEN 2070-2099 wrt. CTRL 1981-2010).

971





**Figure 10** Comparison of temperature probability, snowfall probability and mean snowfall intensity for the CTRL period 1981-2010 and SCEN period 2070-2099 for RCP8.5. The analysis is based on data from the 14 snowfall separated + bias corrected RCM simulations (RCM$_{sep+bc}$). The top row depicts the PDF of the daily temperature distribution, while the second row shows the mean number of snowfall days between September and May, i.e., days with S > 1 mm/d (see Tab. 2), in a particular temperature interval. The third row represents the mean snowfall intensity, S$_{int}$, for a given snowfall temperature intervall. In addition the Clausius-Clapeyron relationship, centred at the -10°C mean S$_{int}$ for SCEN, is displayed by the black dashed line. PDFs and mean S$_{int}$ were calculated by creating daily mean temperature bins of width 1 °C.




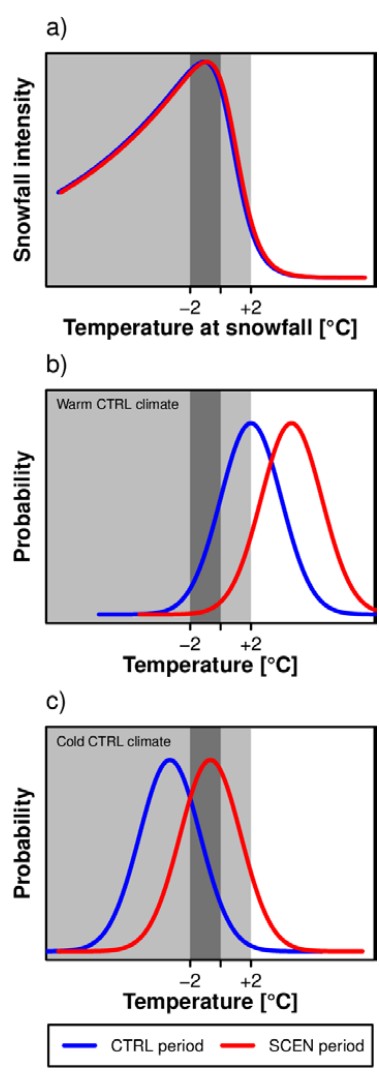

**Figure 11** Schematic illustration of the control of changes in snowfall intensity on changes in mean and extreme snowfall. a) Relation between temperature and mean snowfall intensity. b) Daily temperature PDF for a warm control climate (low elevations or transition seasons). c) Daily temperature PDF for a cold control climate (high elevations or mid-winter). The blue denotes the historical CTRL period, the red line the future SCEN period.





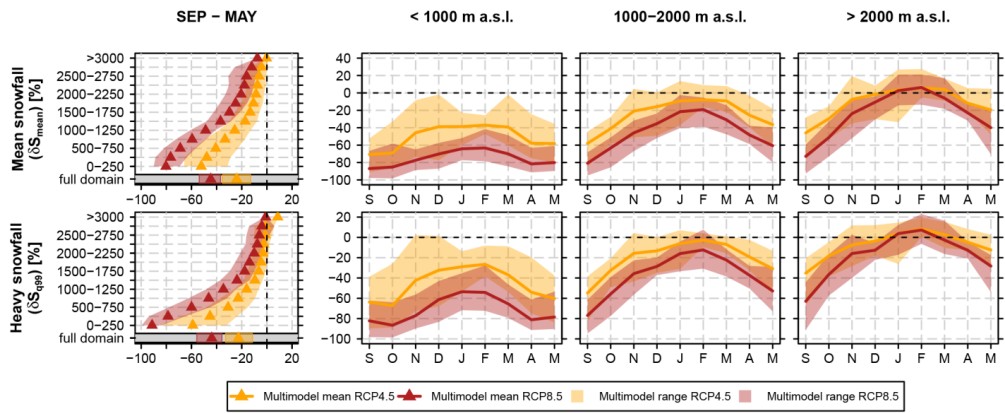


**Figure 12** Similar as Figure 8 but showing projected changes of mean snowfall, $\delta S_{mean}$, and heavy snowfall,
$\delta S_{q99}$, for the emission scenarios RCP4.5 and 8.5. See Fig. S8 for the emission scenario uncertainty of the
remaining four snowfall indices.






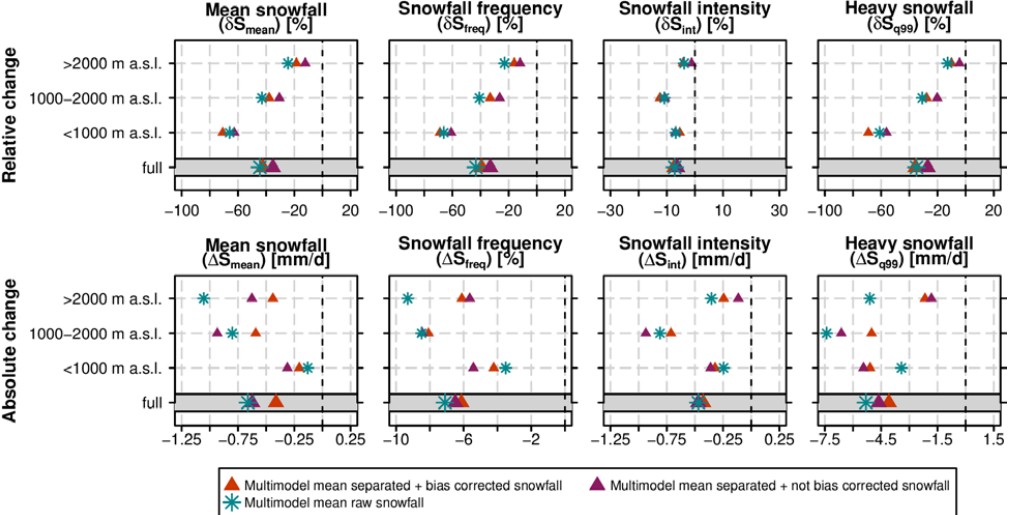



**Figure 13** Relative and absolute changes (SCEN period 2070-2099 with respect to CTRL period 1981-2010) of
mean September-May snowfall indices based on a subset of 9 snowfall separated + bias corrected (RCM$_{sep+bc}$), 9
snowfall separated + not bias corrected (RCM$_{sep+nbc}$) and 9 raw snowfall RCM simulations (RCM$_{raw}$) for RCP8.5.
Only RCM simulations providing raw snowfall as output variable (see Tab. 1) were used in this analysis.





**Tables**

**Table 1** Overview on the 14 EURO-CORDEX simulations available for his study. The whole model set consists of
seven RCMs driven by five different GCMs. All experiments were realized on a grid, covering the European
domain, with a horizontal resolution of approximately 12.5 km (EUR-11) and were run for control RCP4.5 and
RCP8.5 scenarios within the considered time periods of interest. A subset of 9 simulations provides raw snowfall,
i.e., snowfall flux in $kg/m^2 s$, as output variable. For full institutional names the reader is referred to the official
EURO-CORDEX website www.euro-cordex.net.

| RCM | GCM | Acronym | Institute ID | Raw snowfall output |
|-----|-----|---------|--------------|---------------------|
| ALADIN53 | CNRM-CERFACS-CNRM-CM5 | CNRM - ALADIN | CNRM | no |
| CCLM4-8-17 | CNRM-CERFACS-CNRM-CM5 | CNRM - CCLM | CLMcom/BTU | no |
| CCLM4-8-17 | ICHEC-EC-EARTH | EC-EARTH - CCLM | CLMcom/BTU | no |
| CCLM4-8-17 | MOHC-HadGEM2-ES | HadGEM2 - CCLM | CLMcom/ETH | no |
| CCLM4-8-17 | MPI-M-MPI-ESM-LR | MPI-ESM - CCLM | CLMcom/BTU | no |
| HIRHAM5 | ICHEC-EC-EARTH | EC-EARTH - HIRHAM | DMI | yes |
| RACMO22E | ICHEC-EC-EARTH | EC-EARTH - RACMO | KNMI | yes |
| RCA4 | CNRM-CERFACS-CNRM-CM5 | CNRM - RCA | SMHI | yes |
| RCA4 | ICHEC-EC-EARTH | EC-EARTH - RCA | SMHI | yes |
| RCA4 | MOHC-HadGEM2-ES | HadGEM2 - RCA | SMHI | yes |
| RCA4 | IPSL-IPSL-CM5A-MR | IPSL - RCA | SMHI | yes |
| RCA4 | MPI-M-MPI-ESM-LR | MPI-ESM – RCA | SMHI | yes |
| REMO2009 | MPI-M-MPI-ESM-LR | MPI-ESM – REMO* | MPI-CSC | yes |
| WRF331F | IPSL-IPSL-CM5A-MR | IPSL - WRF | IPSL-INERIS | yes |

* r1i1p1 realisation







**Table 2** Analysed snowfall indices. The last column indicates the threshold value in the CTRL period for considering a grid cell in the climate changes analysis (grid cells with smaller values are skipped for the respective analysis); first number: threshold for monthly analyses, second number: threshold for seasonal analysis.

| Index name | Acronym | Unit | Definition | Threshold for monthly / seasonal analysis |
|---|---|---|---|---|
| Mean snowfall | $S_{mean}$ | mm | (Spatio-)temporal mean snowfall in mm snow water equivalent (only "mm" thereafter). | 1 mm / 10 mm |
| Heavy snowfall | $S_{q99}$ | mm/d | Grid point-based 99% all day snowfall percentile. | 1 mm / 1 mm |
| Max. 1 day snowfall | $S_{1d}$ | mm/d | Mean of each season's or month's maximum 1 day snowfall. | 1 mm / 1 mm |
| Snowfall frequency | $S_{freq}$ | % | Percentage of days with snowfall S>1mm/d within a specific time period. | 1 % / 1 % |
| Snowfall intensity | $S_{int}$ | mm/d | Mean snowfall intensity at days with snowfall S>1mm/d within a specific time period. | $S_{freq}$ threshold passed |
| Snowfall fraction | $S_{frac}$ | % | Percentage of total snowfall, $S_{tot}$, on total precipitation, $P_{tot}$, within a specific time period. | 1 % / 1 % |