# Peer review of "Future snowfall in the Alps: Projections based on the EURO-CORDEX regional climate models"

_The Cryosphere, 2017_

## Referee Comment (RC1) · Anonymous Referee #1 · 5 Mar 2017

Review of "Snowfall in the Alps: Evaluation and projections based on the EURO-CORDEX regional climate model" by P. Frei, S. Kotlarski, M. A. Liniger and C. Schär.

Recommendation: acceptation with minor revisions

The authors evaluate snowfall from 14 high-resolution EURO-CORDEX simulations. They use a method to re-calculate snowfall based on near-surface temperature conditions and designed to account for subgrid-scale topography, and a method to correct biases. Interestingly, they also analyse the raw snowfall from EURO-CORDEX simulations when available. They assess the performance of their methods by comparing several snowfall indices to the ones derived from observational products, which is an interesting approach. Then, they consider projections under the RCP85 scenario, and

explain changes in the aforementioned indices.

This is a nice piece of work, the paper is well written and the model analysis is robust. I had several important questions while reading sections 3 and 4, but all of them were addressed in section 5 (Discussion). So I recommend this paper for publication in TC. I just have a bunch of minor comments that are listed below.

Minor comments:

- L. 65-66: replace "the GCM provides the lateral boundary conditions to the RCM" with "the GCM provides the lateral and sea surface boundary conditions to the RCM". - L. 88-89: the fact that "a gridded observational snowfall product that could serve as reference for RCM evaluation does not exist" is not a good reason for not using raw outputs, it can actually be evaluated as in Fig. 13 of this paper... - L. 133-135: the RCMs also have an effective resolution that is larger than their grid resolution, see e.g. Skamarock et al. (Mon. Wea. Rev. 2004), Lefèvre et al. (Mar. Pol. Bull. 2010). - L. 151: the authors should make clear that what they refer to as "control" is based on the CMIP5 historical simulations (not the one based on reanalyses). - Section 2.2: it is worth mentioning that daily-averages from EURO-CORDEX are used (or specify what other time sampling/averaging is used). - Fig. 1: which topography is shown? - L.246-249: it is not clear to me why the explanation cannot be reversed: coarse cells with grid temperature lower than T* should overestimate snowfall at some locations covered by the cell, don't they? If it's not zero on average, is it because elevation distribution is generally skewed within the coarse cell? - L. 463: I think that "Rhone Valley" would be more appropriate than "Western France". - L. 485: typo "change sin". - Fig. 11: indicate what the grey area represents. - Section 5.3: that relative changes in snowfall from raw model outputs are very similar to separated and bias-corrected fields is a very interesting finding and should definitely be reported in the Abstract. - L.656-658: It is indeed a pity that no evaluation is performed based on datasets from other alpine countries. It would also help validate the overall methodology since the methods tuning is undertaken over the Swiss Alps.

---

## Referee Comment (RC2) · Anonymous Referee #2 · 27 Mar 2017

General comments

The paper "Snowfall in the Alps: Evaluation and projections based on the EURO-CORDEX regional climate models" by Prisco Frei, Sven Kotlarski, Mark A. Liniger, Christoph Schär aims to enlight the ability of the EURO-CORDEX models to represent snowfall over the Alpine region in the last decades and to provide future projections for the late 21st century.

The topic addressed by the paper is of interest for a broad community and within the scopes of the Journal. The paper is well written, with a proper language and in good English. The objective of the work clearly stated, the methods, the scientific results and conclusions presented in a clear and concise way.

[Figure]

However, the manuscript presents some inconsistencies in the methodology that needs to be addressed before pubblication.

First of all, in absence of a daily observational gridded snowfall dataset for the Alpine region the authors derive a dataset using daily temperatures and precipitation. They separate the snow fraction from the total precipitation using a fixed temperature threshold $T^*=2°C$. While this method works fine with hourly data, it is probably weak when using data at daily scale. As the rest of the paper builds on the hypotesis that this snowfall dataset represents the ground truth (i.e. the hypothesis is used for the calculation of Richardson snowfall fraction $f_{s,Ri}$ ; for the bias correction of RCM snowfall fields) authors should provide some evidence that their snowfall dataset closely represent the real snowfall distibution.

In the Results (section 3.1) it comes really unexpected that the authors validate the RCM raw snowfall outputs by using 29 fresh-snow daily time series from MeteoSwiss stations. This dataset was not presented before and should be described in the "Observational datasets" section. Moreover this datasets is by definition "the" ground-truth, and I wonder why it comes out only at this point. It should be used for a detailed validation of the 2 km gridded snowfall product that you derive from temperature and precipitation fields. How the 2km gridded product compares to the fresh snow observations? Does it represent properly the snowfall climatology (mean, extremes) in correspondence of the stations? Does it represent the altitudinal gradient of mean/extreme snowfalls intensities? This information on the quality of the gridded reference dataset should be provided as it is necessary to set the basis for the whole methodology.

Finally the RCM bias correction methods is calibrated on the area of Switzerland only and then applied to the whole Alpine region. This is justified by the authors with the lack of information on snowfall beyond the borders of Switzerland. Indeed previous efforts were made to derive a gridded snowfall dataset for the Alpine Region: HISTALP dataset provides monthly snowfall over the full Alpine domain, since 1800, at about 10 km spatial resolution, i.e. resolution comparable to the RCM gridsize (12 km). I

believe this manuscript should include the HISTALP dataset in the analysis, in order to provide a comprehensive view on the reference datasets. In particular I would suggest to discuss i) how HISTALP compares with the stations and the 2km gridded dataset in the Swiss Alps; ii) if it is a good quality reference for validating the RCM snowfall outputs at monthly (or longer) time scales over the Swiss domain.

Specific comments

- P 7 L254: "We apply this regression to relate the surface temperature T to the snow fraction fs by accounting for the topographic subgrid variability. At each coarse grid-point k, the Richards method-based snowfall fraction fs,RI for a given day is hence computed as follows ... First, we estimate the two parameters C and D of Equation 4 for each single coarse grid point k by minimizing the least-square distance to the fs values derived by the Subgrid method via the reference snowfall SSG (local fit)."

The method used to separate snowfall with the temperature threshold T*=2°C is effective with hourly data but it is crude when using daily data as it returns snowfall fraction fs=1 or fs=0 in a given day. This can be far from the reality, expecially at middle elevations (throughout the snow season) but also at high elevations in spring and autumn. The fs,RI depends on the C and D, and the latter are estimated assuming that fs is a good estimator of the solid precipitation fraction. But as said above fs is characterized by unknown uncertainty. You should prove that fs closely reproduce the real snowfall fraction, before applying your method for deriving fs,RI and your snowfall reference dataset. Minimum requirement is to provide a quantification of this error, using fresh snow manual observations in the 29 manual stations.

- P9 L317-321: "the initial snow fractionation temperature T*=2°C of the Richards separation method (see Sec 2.5) is shifted to the value T*a for which the spatially and temporally averaged simulated snowfall amounts for elevations below 2750 m a.s.l. match the respective observation-based reference."

With this temperature correction you basically report the RCM snowfall to your reference. So, also in this case, before applying this procedure you need an evaluation of the error on your snowfall reference. Moreover, can you give details on how you calculate spatial and temporal averages, i.e. which domain/ time range?

- P9 L327: "Note that, as the underlying high-resolution data sets are available over Switzerland only, the calibration of the bias correction methodology is correspondingly restricted, but the correction is then applied to the whole Alpine domain."

HISTALP dataset provides gridded snowfall monthly fields for the Alpine region at about 10 km spatial resolution (Chimani et al 2011), it should be included in the analysis and discussed in comparison to your reference/manual observations in Switzerland

Chimani, B., Böhm R., Matulla C., Ganekind M.: Development of a longterm dataset of solid/liquid precipitation Adv.Sci.Res,6,39-43, 2011http://www.adv-sci-res.net/6/39/2011/asr-6-39-2011.html

- P10 L338-341: "EURO-CORDEX simulations ... are compared against observations derived from measured fresh snow sums from 29 Meteo Swiss stations with data available for at least 80% ofÂăthe evaluation period. For this purpose a mean snow density of 100 kg/m3 for the conversion from measured snow height to water equivalent is assumed."

As said before, I am surprised to see at this point of the paper that you have 29 fresh snowfall time series covering the 1970-2005 period. They should be presented before (section 3.1) & exploited much more than you do. These manned observations are the ground truth and they should be used to validate the snowfall gridded dataset that you derive from temperature and precipitation over Switzerland. Please provide a quality control of the snowfall gridded dataset prior to use it

- P10 L345-347: "The positive bias at high elevations might arise from the factÂăthat the very few observations were made at a specific location while simulated grid point values ofÂăthe corresponding elevation interval might be located in different areas of

Switzerland."

Here you consider all Switzerland and you really mix very different areas far away one from another. Please discuss the case when only the gridpoints containing stations are considered, i.e. showing the spread of the models around the observed time series (i.e. in plots for the three - low, middle and high - elevation ranges?

- P14 L487: "In between is a transition zone with rather strong changes with elevation" . Can you explain why?

- P14 L494-6: Could it also be residual biases along the snowfall line?

- P16 L587-8: Given more precipitation at high elevation & temperatures more favorable to heavy snowfalls, why does the snowfall frequency decrease?

Technical corrections

- P2, L42: Climate models do not "simulate the anthropogenic greenhouse effect", but the provide projections of the future climate in different scenarios, i.e. assuming different of rates of emissions of the greenhouse gases, land cover/use changes .. This sentence should be rephrased with more proper terminology

- P2 L50: "Although the snowfall fraction is expected to decrease at lower elevations during the 21st century": here one or more citations is needed

- P2 L57: "Projections of future changes in theÂăsnowfall climate" -> "Projections of future changes inÂăsnowfall"

- P2 L63: I suggest to remove "of the projections"

- P3 L79: "Low"->"Lower"

- P3 L87: "coarse resolution RCM grid" : here "coarse resolution" is misleading as in the former lines you consider RCM resolution as "high". I suggest to remove "coarse resolution"

- P3 L89: "a gridded observational snowfall product that could serve as reference for RCM evaluation does not exist". This is true only for data at daily scale (see comment above on HISTALP)

- P3 L91: "Air" temperature

- P3 L94: "on the coarser (RCM) grid" -> "on the coarser RCM grid"

- P3 L113: "potentially leading to a reduction in overall snowfall amounts changes": "changes" should be removed

- P4 L115: "here events > 1 mm/day": please give a more a accurate definition of events, I assume "P>1 mm/day"

- P4 Section 2.1: Can you provide some information on the stations included in the two gridded datasets (precipitation & temperatures), i.e. how many are they? Are the same stations measuring both variables?

- P4 L139: "errors might be induced by . . ." I would also mention the uneven distribution of the stations and the under-representation of high altitudes.

- P4 L142: "in innerÂăAlpine valleys, where the presence of cold air pools is systematically overestimated" please provide a reference

- P5 L157: "The six RCM considered" -> They should be seven

- P6 211: "This method also allows for a more physically-based bias correction of simulated snowfall amounts (see Sec. 2.6)." This sentence is unclear here. I would move at the end of the paragraph: ". . . traditional bias correction approaches based only on a comparison of observed and simulated snowfall amounts in the historical climate would possibly fail due to a non-stationary bias structure, while the present method also allows for a more physically-based bias correction of simulated snowfall Âăamounts (see Sec. 2.6)."

- P7 L224 "coarse grid information" -> "fine grid information"?

- P7 L258 The two parameters C and D are defined as "point of inflection" and "growth rate" without giving other details. Please provide an explanation of the physical meaning.

- P8 L280: "For both indices Smean and Sq99, mean ratios across all elevation intervals are close to 1 (Fig. 2). At single grid points, maximum deviations are not larger than 1±0.1." At single gridpoints the spread seems ±0.3 for the binary method (Fig 2)

- P9 L316: "Pcorr approximately corresponds to the observation-based estimate" I would add "in the evaluation period"

- P10 L362 Fig. S1 -> Fig.S2

- P11 L372-377 : Can you provide more detail on the extent of the undercatch?

- P11 L388: "Bias of close to" -> Bias close to

- P13 L471 : between "effect" and "of" it seems there are too many spaces?

- P13 L485: "change sin" -> "changes in";

- P13 L486: "in the upper part" -> "at higher elevation"

- P19 L692: ".."

- P19 L700: "snowfall-friendly temperature range" : Nice however I would say "temperature range more favorable to snowfall"

- Figure 1: What is the souce for this topography?

- Figure 6: The differences among these plots are not easy to see, I would recommend to show biases with respect to the reference.

- Figure 11: How low and high elevations are defined? Transition seasons are MAM and SON?

- Table 1: Caption "his study" -> "this study". Moreover, please check the "*r1i1p1"

below the table, which models does it refer to?

---

## Referee Comment (RC3) · Anonymous Referee #3 · 5 Apr 2017

MS No.: tc-2017-7 Review, 5 April 2017

Title: Snowfall in the Alps: Evaluation and projections based on the EURO-CORDEX regional climate models

Authors: Prisco Frei, Sven Kotlarski, Mark A. Liniger, Christoph Schär

Recommendation: [Revision]

GENERAL COMMENTS:

This article assesses the performance of (an ensemble of) simulations with regional climate models conducted in the framework of EURO-CORDEX in representing snowfall

characteristics in the Alps and the surrounding regions. It also discusses the changes in snowfall projected by the model ensemble under the RCP8.5 and 4.5 emission scenarios. Rather than taking snowfall from the raw (or native) model output (which is not available for all RCMs) is derives snowfall amount from total precipitation and near surface temperature. The newly develop method also accounts for the effect of subgrid-scale orographic variance. The authors show subsequently that a form of bias adjustment in precipitation and temperature for each GCM-RCM member separately is needed to allow a meaningful comparison with an 'observed' reference state.

Overall the paper is coherently written, and the methodology applied to bring model output from the various GCM-RCM simulations in a common framework regarding snowfall characteristics is adequately presented and properly motivated. The work is not entirely novel, many of the findings confirm conclusions from previous studies. The strength of the paper is that the authors developed a well-structured approach to assess the performance of a variety of models, each with its own characteristics, in representing a relevant climate parameter which is difficult to access through direct observations.

Having said that, I think the authors have still a bit of work to do. Throughout the paper I came across a number of issues which require some further attention from the authors before the paper can be accepted. Yet, I am sure that after the authors have adequately addressed my points of concern a revised manuscript will be suitable for publication.

MAJOR COMMENTS:

1. Half-way the introduction (lines 75-80) the authors write " Within the last few years . . ." followed by "Most of these analyses are based on GCM output or older generations of RCM ensembles at comparatively low spatial resolution . . .". This may indeed be the case for most of the studies cited in the sentence before, but not for all of them, and the authors should specify explicitly which of them is the exception, and how that study

compares with their work. E.g. Piazza et al. use, amongst other models, a number of RCMs operated at 12 km resolution, the study by de Vries et al. (2014) is based on a 8-member ensemble of EC-EARTH-RACMO simulations at 12km resolution (historical and rcp8.5) configured on a smaller domain, but in principle quite comparable to the simulations used in this paper.

2. Instead of using "bias correction" I would strongly recommend to use the phrase "bias adjustment" as was also the adopted terminology by the EURO-CORDEX community. The word "correction" suggests there is a well-established methodology including a ground truth observed state which we all agree on. This is obviously not the case. It is therefore much better to use the word "adjustment" which automatically triggers the questions "how" and "to what" or "in which context" as it should be.

3. In section 2.2 it is mentioned that all GCM-driven EUR-11 simulations for which control, RCP4.5 and RCP8.5 runs are currently available have been included in the study. This can obviously not be a correct state of affairs. Currently means "at the moment" and since the number of simulations published in the ESGF-archive is still growing, there will be a moment that the statement is no longer true. In fact, already in October 2016 there were 16 simulations that met the criteria set by the authors. In addition to the their selection there were results published from HadGEM2-RACMO and MPI-ESM (r2i1p1)-REMO. In April 2016, none of the two MPI-ESM-REMO simulations were available, but the HadGEM2-RACMO simulation was, albeit based on version v1 which was replaced by version v2 in August 2016. So, a) you need to specify currently, and b), either include the simulations that were in the archive but not in your selection, or convincingly explain why some simulations were not selected.

4. Following the previous point, it is important mentioning that three different realizations of ICHEC-EC-EARTH are used to force four different RCMs: r12i1p1 is used to force CCLM and RCA, r3i1p1 is used for HIRHAM, and r1i1p1 for RACMO. Different realizations of a GCM can show distinct behavior owing to long-term large-scale natural variability implying that differences between the EC-EARTH forced RCM simulations

are not only due to differences in the RCMs. This would only hold for the CCLM end RCA simulations forced with r12i1p1. Please, mention this aspect when introducing the 14 GCM-RCM chains.

5. In section 2.5 the methodology to separate snowfall from total precipitation is discussed (Richards method). In the final paragraph a parametric formulation $f_{s,Ri}$ is introduced (Eqs 4-7) to express the snowfall fraction in terms of coarse-grid temperature $T_k$, the topographic standard deviation $\sigma_h$ of the designated grid cell, and a number of constants (E,F,G,H) which are determined through an empirical fitting procedure. The function $f_{s,Ri}$ is meant to be used to separate snowfall in the RCMs as well (line 282-283). Since the subgrid-scale orographic variance parameter of the model orography is not known (at least this parameter is not in the ESGF-archive) I presume the authors have used $\sigma_h$ from the observational dataset. Somehow the observational $\sigma_h$ and model height should match. This is not guaranteed a priori. The authors already mention that orography fields from different RCMs can be quite different from each other, and from the observations (line 157-159). The authors should explain how they deal with such mismatches.

6. In section 2.6 the bias adjustment approach is discussed. I was surprised to see that after the detailed treatment of snowfall separation, the adjustment of temperature has been dealt with so crudely. While according to Fig S4 the temperature bias considerably depends on elevation the authors have chosen the shifted fractionation temperature to be independent of elevation. According to Fig 4 the adjusted fractionation temperature and the temperature bias (one point per GCM-RCM realization) show considerable scatter around a linear relation, but it is not at all clear and also not explained what causes this scatter. It might be due to bias depending to elevation, but also to month of the year and/or region. Or is there something else? The authors should discuss their treatment of shifting the fractionation temperature in more detail, it is particularly relevant because the snow fractionation temperature appears to be a crucial, and probably also a sensitive, parameter in the analysis.

7. Line 565-566. The sentence "Previous studies . . . with this theory (e.g. Allen and Ingram, 2002; Ban et al. 2015)" is completely out of context. I strongly suggest to omit this sentence and the corresponding references. The focus of the Ban et al. paper is on summertime convectively driven sub-daily (hourly) precipitation extremes and its relation with temperature. This is miles away from the Sq99 and S1d snowfall parameters used in this paper to indicate heavy (but not extreme) snowfall at the daily scale outside the summer season.

MINOR COMMENTS:

1. In the introduction line 46-47: "First, total snowfall sums are expected to decrease by a decreasing probability for precipitation to fall as snow and a decreasing snowfall fraction (ratio between solid and total precipitation)." I do not see the difference between the two reasons, I would say that the first reason implies the second. Suggest to retain only the first part.

2. Line 107: rephrase "On centennial time scales" as "On decadal to centennial time scales"

3. Line 157 reads "It is important to note that each of the six RCMS considered . . ." I presume six must be seven in accordance with Table 1 and line 152.

4. Throughout the text subgrid should be rephrased as subgrid-scale. (that is apart from the few places where scale was already appended)

5. Line 174: "present day" → " present-day"

6. Line 196: "were initially set". What is the meaning of "initially". I suggest to write just "are set"

7. Line 203. "possible" → "possibly"

8. Line 245-246: According to the text "the Binary method underestimates Smean and overestimates Sq99 for all elevation intervals (Fig.2.)". Looking at Fig 2 this is not true

for Smean at the highest elevations. Adapt text accordingly.

9. Connected to the previous point, in caption Fig 2 the text reads "... and the full subgrid snow representation" I presume this is the same as the reference snowfall SSG which is used in the main text. If correct, use same terminology in caption as in main text, if not correct, explain the meaning of full subgrid-scale snow representation ?

10. Line 247: "... even for coarse grid temperature ..."→"... even for a coarse grid temperature ..." or "... even for coarse grid temperatures ..."

11. Line 258: "with C the point of inflexion, and the growth rate D." →" with C the point of inflexion, and D the growth rate."

12. In line 339 the text mentions "250 m-elevation intervals in the range 950-1650 m". I don't see how 250-m intervals match in the range 950-1650 m" Please explain.

13. Line 362: "... elevations (Fig. S1)." Presume this must refer to Fig. S2.

14. Line 364:"...observations approximately linearly depends on elevation" rephrase as "... observations depends approximately linearly on elevation"

15. Line 376-377: "However, a potential undercatch can only partly explain the overestimation of precipitation found in the present work." Please, discuss why it can only be a partly explanation, and not the full explanation

16. Line 430: "This again highlights the fact ...". highlights is too strong a word. Suggest to use illustrates.

17. Line 468 (connects to main point 4). "(e.g. the four simulations driven by EC-EARTH)" As pointed out before, the EC-EARTH forcings were inferred from three different realizations.

18. Line 473: ".. in Fig.8 that ..." → " .. in Fig. 8 which ..."

Interactive
comment

19. Line 483: rephrase "... which forces the multimodel mean change to be slightly positive in January ..." as " ... which leads to a slightly positive change in multi-model mean for January ..."

20. Line 485: "... change sin ..." → " ... changes in..."

21. Line 506: "... mutlimodel ..." → " ... multi-model ..." (multimodel requires a hyphen everywhere)

22. Line 534: " ... and > 2000," → "... and > 2000 m a.s.l."

23. Line 595: rephrase "This is approximately given ..." as "This behavior is approximately found ..."

24. Line 639-640: "(which is for many GCM-RCM chains not available)" For most model chains it is available so many is a too strong qualification. Suggest instead "(which is not available for all GCM-RCM chains)"

25. Line 690: "We here confirm this conclusion." Too strong phrasing, please write "We here confirm this finding."

26. Line 690-691: rephrase " ... the optimal temperature for heavy snowfall will still occur in a warmer climate and, hence, ..." as " ...heavy snowfall in a warmer climate will still occur in the optimal temperature range, hence, ..."

27. Line 700: "... snowfall-friendly temperature range ..." Funny wording!

28. It would help if e.g. Fig 3a contains an ancillary line indicating the relative occurrence of the 250-m elevation ranges for Swiss and/or the Alpine region (based on the topography map in Fig 1) with labels on the upper horizontal axis.

29. The colour scheme used in Figs 7 and S5,S6,S7 is lacking contrast. Would it be possible to use a wider and somewhat lighter colour scheme?

30. Caption Fig 2: "factionation"→"fractionation"

---

## Author Comment (AC1) · 30 May 2017

**Future snowfall in the Alps: Projections based on the EURO-CORDEX regional climate models**

Prisco Frei, Sven Kotlarski, Mark A. Liniger, Christoph Schär

**- Response to Referees –**

**General**

We thank the three referees for their careful revision of the manuscript and for their constructive comments. Please find below our replies to all major comments and our suggestions on how to address these issues in a revised manuscript. We hope that we satisfactorily addressed all referee comments and that the proposed changes are considered as being appropriate. In case that not, we'd be looking forward to discuss individual remaining issues in more detail.

As several referee comments addressed the RCM evaluation and the evaluation of the 2 km snowfall reference itself, we'd like to put in front the following two statements on the scope of the paper:

(1) Our work is primarily concerned with the analysis of future snowfall projections. However, a basic notion on the quality of raw RCM snowfall and, hence, the general ability of RCMs to represent our variable of interest is required for such an exercise in our opinion. In the manuscript this is accomplished by comparing RCM raw snowfall against site-scale measurements obtained from new snow sums (Figure 3). Such a comparison is subject to considerable uncertainties, mostly originating from the scale gap between RCM grid cells and site-scale observations and from representativity issues of observed snow cover. Due to a missing high-quality observational reference at the scale of the RCM resolution (in our opinion also the HISTALP dataset has its shortcomings; see below) we refrain from evaluating RCM snowfall in more detail and, at least when interpreting raw snowfall change signals, implicitly assume stationary model biases. As a consequence, the projection aspect of the current work is much larger than the evaluation aspect, and we tried to better clarify this issue by modifying the text in Chapters 1 and 3.1. Furthermore, we adjusted the title of the manuscript accordingly and removed the term "Evaluation".

(2) Relating to the previous issue but especially to the validation of the snowfall reference against which the RCM-derived snowfall is adjusted: As it was already mentioned in the introduction of the original manuscript, we do not claim to present an ultimate solution for bias-adjusting RCM-based snowfall but employ a spatially and temporally aggregated adjustment procedure that does nevertheless separately account for temperature and precipitation biases. Aspects of the snowfall climate that are not corrected for, such as details of the spatial snowfall pattern, are described in Sections 2.6 and 3.3. The simplifications also include the fact that we basically accept a non-perfect observation-based reference. A well-validated and appropriate reference does not exist in our opinion (see also above). The very core of our work is the analysis of projected future snowfall changes and the comparison of three different ways to produce such estimates: (1) Raw RCM snowfall, (2) RCM snowfall as separated from simulated temperature and precipitation, and (3) RCM snowfall as separated from simulated temperature and precipitation and additionally bias-adjusted. The latter version is the basic dataset for the climate scenario analysis as it can be constructed for all participating RCMs (raw snowfall is not available for all of them) and as it is, in principle, able to account for temperature-dependent and hence non-stationary snowfall biases. However, Chapter 5.3 shows that relative change estimates largely agree among all three datasets and are robust. From that point of view, the influence of remaining inaccuracies in the bias-adjusted snowfall projections due to inaccuracies of the reference is presumably small. In the revised version we now try to better clarify these issues by modifying Chapters 1, 2.5, 2.6 and 5.3.

**Response to Referee #1**

**Comment** *L. 65-66: replace "the GCM provides the lateral boundary conditions to the RCM" with "the GCM provides the lateral and sea surface boundary conditions to the RCM".*

**Response and changes to manuscript** Modified accordingly.

**Comment** *L. 88-89: the fact that "a gridded observational snowfall product that could serve as reference for RCM evaluation does not exist" is not a good reason for not using raw outputs, it can actually be evaluated as in Fig. 13 of this paper:*

**Response and changes to manuscript** You are perfectly right, thanks very much for pointing this out. The reason is indeed the non-availability of raw snowfall output in several experiments. We removed the second part of this sentence.

**Comment** *L. 133-135: the RCMs also have an effective resolution that is larger than their grid resolution, see e.g. Skamarock et al. (Mon. Wea. Rev. 2004), Lefèvre et al. (Mar. Pol. Bull. 2010).*

**Response and changes to manuscript** Thanks for pointing this out. We now use the term "nominal resolution of the available climate model data", but would like to refrain from further discussing the even coarser effective resolution of climate models as we believe that this would distract the reader at this point.

**Comment** *L. 151: the authors should make clear that what they refer to as "control" is based on the CMIP5 historical simulations (not the one based on reanalyses).*

**Response and changes to manuscript** The fact that GCM-driven experiments are employed has already been mentioned several times in the respective paragraph. We slightly revised this paragraph now to make this point even clearer.

**Comment** *Section 2.2: it is worth mentioning that daily-averages from EURO-CORDEX are used (or specify what other time sampling/averaging is used).*

**Response and changes to manuscript** This point is mentioned now.

**Comment** *Fig. 1: which topography is shown?*

**Response and changes to manuscript** Thanks for this comment, this information was indeed missing so far. The topography shown is the GTOPO30 digital elevation model of the U.S. Geological Survey. The figure caption was adjusted accordingly. As GTOPO30 is also the basis for computing the topographical standard deviation $\sigma_h$ for each RCM grid cell in the course of calibrating the Richards method we furthermore provide this information now in Section 2.5 of the manuscript and also added one sentence to the acknowledgments.

**Comment** *L.246-249: it is not clear to me why the explanation cannot be reversed: coarse cells with grid temperature lower than T\* should overestimate snowfall at some locations covered by the cell, don't they? If it's not zero on average, is it because elevation distribution is generally skewed within the coarse cell?*

**Response and changes to manuscript** A skewed subgrid topography distribution might be one reason, but the main factor is probably the fact that precipitation-elevation gradients over many parts of the analysis domain are positive, i.e. higher total precipitation sums at higher (=colder) elevations. Snowfall separation on the high-resolution grid would therefore lead to higher spatially-averaged mean snowfall sums compared to the coarse-resolution version, hence a systematic non-zero difference of the two versions. We modified the respective text section in order to better clarify this point.

**Comment** *L. 463: I think that "Rhone Valley" would be more appropriate than "Western France".*

**Response and changes to manuscript** Thanks a lot, we modified this sentence accordingly.

**Comment** *L. 485: typo "change sin".*

**Response and changes to manuscript** Corrected.

**Comment** *Fig. 11: indicate what the grey area represents.*

**Response and changes to manuscript** We're sorry, this information was mentioned in the text but not in the figure caption so far. The grey area represents the overall temperature interval at which snowfall occurs (light grey) as well as the preferred temperature interval for heavy snowfall to occur (dark grey). This information has been added to the figure caption now.

**Comment** *Section 5.3: that relative changes in snowfall from raw model outputs are very similar to separated and bias-corrected fields is a very interesting finding and should definitely be reported in the Abstract.*

**Response and changes to manuscript** You are perfectly right, thanks for pointing this out. One sentence on this finding has now been added to the abstract.

**Comment** *L.656-658: It is indeed a pity that no evaluation is performed based on datasets from other alpine countries. It would also help validate the overall methodology since the methods tuning is undertaken over the Swiss Alps.*

**Response and changes to manuscript** Following a suggestion of Referee #2 an additional comparison of the constructed reference snowfall against the HISTALP dataset has been added to the manuscript. Please see our replies to Referee #2 and Figure S5 of the revised manuscript. Please also see our general replies above concerning the importance of an accurate reference in the context of our study.

**Further changes to the manuscript**

**Chapter 2.1 "Observational data"** For reasons of completeness we additionally included the information that the temperature and precipitation grids employed are slightly shifted with respect to their reference time interval (midnight UTC - midnight UTC for temperature, 06 UTC - 06 UTC for precipitation).

**Chapter 2.2 "Climate model data"** In the last paragraph we erroneously spoke of *six* RCMs considered. We corrected this to *seven* RCMs.

**Chapter 3.1** To better account for uncertainties in this simplified evaluation we now additionally cite the work of Grünewald and Lehning (2015) that highlights the danger of non-representativity of single-site snow depth observations in Alpine terrain.

**Figure 3** In the left panel two of the 29 stations employed (WFJ and MVE) were plotted at a wrong elevation in the original version. For both stations the correct elevation differs by about 100 m from the previously used elevation. The figure has been corrected accordingly, in addition to the modifications to this figure mentioned above. The conclusions of the analysis do not change.

**Overall manuscript** Several minor spelling and wording mistakes as well as an inconsistent use of past and present tense were corrected.

[revised manuscript text omitted]

---

## Author Comment (AC2) · 30 May 2017

**Future snowfall in the Alps: Projections based on the EURO-CORDEX regional climate models**

Prisco Frei, Sven Kotlarski, Mark A. Liniger, Christoph Schär

**- Response to Referees –**

**General**

We thank the three referees for their careful revision of the manuscript and for their constructive comments. Please find below our replies to all major comments and our suggestions on how to address these issues in a revised manuscript. We hope that we satisfactorily addressed all referee comments and that the proposed changes are considered as being appropriate. In case that not, we'd be looking forward to discuss individual remaining issues in more detail.

As several referee comments addressed the RCM evaluation and the evaluation of the 2 km snowfall reference itself, we'd like to put in front the following two statements on the scope of the paper:

(1) Our work is primarily concerned with the analysis of future snowfall projections. However, a basic notion on the quality of raw RCM snowfall and, hence, the general ability of RCMs to represent our variable of interest is required for such an exercise in our opinion. In the manuscript this is accomplished by comparing RCM raw snowfall against site-scale measurements obtained from new snow sums (Figure 3). Such a comparison is subject to considerable uncertainties, mostly originating from the scale gap between RCM grid cells and site-scale observations and from representativity issues of observed snow cover. Due to a missing high-quality observational reference at the scale of the RCM resolution (in our opinion also the HISTALP dataset has its shortcomings; see below) we refrain from evaluating RCM snowfall in more detail and, at least when interpreting raw snowfall change signals, implicitly assume stationary model biases. As a consequence, the projection aspect of the current work is much larger than the evaluation aspect, and we tried to better clarify this issue by modifying the text in Chapters 1 and 3.1. Furthermore, we adjusted the title of the manuscript accordingly and removed the term "Evaluation".

(2) Relating to the previous issue but especially to the validation of the snowfall reference against which the RCM-derived snowfall is adjusted: As it was already mentioned in the introduction of the original manuscript, we do not claim to present an ultimate solution for bias-adjusting RCM-based snowfall but employ a spatially and temporally aggregated adjustment procedure that does nevertheless separately account for temperature and precipitation biases. Aspects of the snowfall climate that are not corrected for, such as details of the spatial snowfall pattern, are described in Sections 2.6 and 3.3. The simplifications also include the fact that we basically accept a non-perfect observation-based reference. A well-validated and appropriate reference does not exist in our opinion (see also above). The very core of our work is the analysis of projected future snowfall changes and the comparison of three different ways to produce such estimates: (1) Raw RCM snowfall, (2) RCM snowfall as separated from simulated temperature and precipitation, and (3) RCM snowfall as separated from simulated temperature and precipitation and additionally bias-adjusted. The latter version is the basic dataset for the climate scenario analysis as it can be constructed for all participating RCMs (raw snowfall is not available for all of them) and as it is, in principle, able to account for temperature-dependent and hence non-stationary snowfall biases. However, Chapter 5.3 shows that relative change estimates largely agree among all three datasets and are robust. From that point of view, the influence of remaining inaccuracies in the bias-adjusted snowfall projections due to inaccuracies of the reference is presumably small. In the revised version we now try to better clarify these issues by modifying Chapters 1, 2.5, 2.6 and 5.3.

**Response to Referee #2**

**Comment** *First of all, in absence of a daily observational gridded snowfall dataset for the Alpine region the authors derive a dataset using daily temperatures and precipitation. They separate the snow fraction from the total precipitation using a fixed temperature threshold T\*=2 C. While this method works fine with hourly data, it is probably weak when using data at daily scale. As the rest of the paper builds on the hypothesis that this snowfall dataset represents the ground truth (i.e. the hypothesis is used for the calculation of Richardson snowfall fraction fs,Ri ; for the bias correction of RCM snowfall fields) authors should provide some evidence that their snowfall dataset closely represent the real snowfall distribution.*

**Response and changes to manuscript** We are thankful for this comment. It relates to a detail of the manuscript where (1) we might have been misunderstood when laying out the scope and the objectives of the work, but where also (2) a proper comparison to other datasets was obviously missing. Please see our comments in the introduction of these replies concerning the scope and objectives of the work. In the revised manuscript we now try to make the point clear that our climate scenario assessment is, in the end, based on three different snowfall datasets that differ with respect to if and how the climate model data were postprocessed. Chapter 5.3 of the manuscript inter-compares the three approaches and concludes that at least for relative change signals the results are robust and do only slightly depend on the postprocessing strategy that is applied. The observation-based snowfall reference grid is used in two of these approaches, but not in the assessment based on raw RCM snowfall. The fact that all three estimates basically agree with each other in terms of relative change signals (which are the core of the paper) downweights the importance of the reference dataset for snowfall separation and bias-adjustment, and we can accept an only approximate reproduction of the (unknown) real snowfall climate. Concerning the evaluation of the reference snowfall grid the new manuscript version now includes a comparison to station-based fresh snow sums as well as a comparison to the HISTALP product. These additional analyses are part of the new sub-Chapter 3.2 "Evaluation of the reference snowfall". Please see the replies below for further details. We hope these changes to the manuscript are considered appropriate.

**Comment** *In the Results (section 3.1) it comes really unexpected that the authors validate the RCM raw snowfall outputs by using 29 fresh-snow daily time series from MeteoSwiss stations. This dataset was not presented before and should be described in the "Observational datasets" section. Moreover this datasets is by definition "the" ground-truth, and I wonder why it comes out only at this point. It should be used for a detailed validation of the 2 km gridded snowfall product that you derive from temperature and precipitation fields. How the 2km gridded product compares to the fresh snow observations? Does it represent properly the snowfall climatology (mean, extremes) in correspondence of the stations? Does it represent the altitudinal gradient of mean/extreme snowfalls intensities? This information on the quality of the gridded reference dataset should be provided as it is necessary to set the basis for the whole methodology.*

**and**

**Comment** *Finally the RCM bias correction methods is calibrated on the area of Switzerland only and then applied to the whole Alpine region. This is justified by the authors with the lack of information on snowfall beyond the borders of Switzerland. Indeed previous efforts were made to derive a gridded snowfall dataset for the Alpine Region: HISTALP dataset provides monthly snowfall over the full Alpine domain, since 1800, at about 10 km spatial resolution, i.e. resolution comparable to the RCM gridsize (12 km). I believe this manuscript should include the HISTALP dataset in the analysis, in order to provide a comprehensive view on the reference datasets. In particular I would suggest to discuss i) how HISTALP compares with the stations and the 2km gridded dataset in the Swiss Alps; ii) if it is a good quality reference for validating the RCM snowfall outputs at monthly (or longer) time scales over the Swiss domain.*

**Response and changes to manuscript** Thank you very much for these detailed suggestions on improving the manuscript. We agree that a quality assessment of our 2 km snowfall reference has been missing so far. At the same time, however, a comparison against the station-based fresh snow sums is subject to considerable uncertainties as well (see our comments in the beginning of these replies and the new text section in Chapter 3.1). Furthermore, the bias-adjusted RCM snowfall (adjusted against the aggregated 2 km reference) is only one out of three estimates used for the snowfall projections. The three estimates yield rather similar results in terms of relative snowfall changes, which downweights the relevance of the specific reference used.

Altogether, we still agree that some quality assessment of the reference is helpful. For this purpose we introduced a new sub-Chapter 3.2 "Evaluation of reference snowfall". This sub-Chapter includes a modification to the existing Figure 3 (additional comparison of the reference against the new snow observations at stations in terms of the mean snowfall climate) and a new supplementary Figure S5 (comparison against the monthly HISTALP dataset). Regarding the suggested evaluation of individual grid cells of the 2km gridded snowfall reference against the fresh snow sums at stations we refrain from including this analysis at a prominent place in the paper as little can be learned due to the remaining scale gap (2 km grid cells vs. site scale) and the problem of non-representativity of snow depth measurements in topographically structured terrain (see new text in Chapter 3.1). For your information, we however present this analysis in Figure R1 below. Despite the inherent uncertainties of the intercomparison, the 2 km snowfall reference and the site observations agree fairly well with each other in terms of climatological mean snowfall. This basic information is now also provided in the manuscript (Chapter 3.2) though without showing the figure.

[Figure]

**Figure R1**: As Figure 3 of the revised manuscript but for the simulated data (green) and for the 2 km snowfall reference (black)
only those grid cells that directly cover the 29 MeteoSwiss stations are considered.

Concerning the evaluation against HISTALP, a comparison of the 2 km reference as well as of the reference on
the RCM grid (after applying the Richards equation) yields an approximate agreement. However, due to the
method used to construct the HISTALP solid precipitation grid (application of monthly snowfall fraction factors to a
spatially interpolated total precipitation grid) and the comparatively coarse station network considered for the total
precipitation grid (192 stations) we believe that also the HISTALP reference is subject to considerable
uncertainties and only a qualitative comparison is valid. The strength of HISTALP clearly lies in the long period
covered, but not necessarily in spatial detail at high (daily) temporal resolution.

Following your suggestion, we now also introduce the station-based fresh snow sum dataset in Chapter 2.1.

**Comment** *P 7 L254: "We apply this regression to relate the surface temperature T to the snow fraction fs by accounting for the topographic*
*subgrid variability. At each coarse gridpoint k, the Richards method-based snowfall fraction fs,RI for a given day is hence computed as follows ...*
*First, we estimate the two parameters C and D of Equation 4 for each single coarse grid point k by minimizing the least-square distance to the fs*
*values derived by the Subgrid method via the reference snowfall SSG (local fit)."*

*The method used to separate snowfall with the temperature threshold T\*=2 C is effective with hourly data but it is crude when using daily data as it*
*returns snowfall fraction fs=1 or fs=0 in a given day. This can be far from the reality, especially at middle elevations (throughout the snow season)*
*but also at high elevations in spring and autumn. The fs,RI depends on the C and D, and the latter are estimated assuming that fs is a good*
*estimator of the solid precipitation fraction. But as said above fs is characterized by unknown uncertainty. You should prove that fs closely*
*reproduce the real snowfall fraction, before applying your method for deriving fs,RI and your snowfall reference dataset. Minimum requirement is*
*to provide a quantification of this error, using fresh snow manual observations in the 29 manual stations.*

**Response and changes to manuscript** Please see our replies above and at the beginning of the replies section.
The 2 km reference snowfall grid is now approximately evaluated. However, your concerns are certainly right. Any
binary method based on near-surface air temperature can only be an approximation of true snowfall. further
presumably more accurate methods are listed by Steinacker (2013). But note that binary methods on daily scales
are frequently used in the literature to separate snowfall from total precipitation, for instance in the hydrological
and glaciological modelling community. A further example cited in the manuscript is the work of Zubler et al.
(2014) who applied the same binary separation at a temperature threshold of 2°C.

**Comment** *P9 L317-321: "the initial snow fractionation temperature T\*=2 C of the Richards separation method (see Sec 2.5) is shifted to the value*
*T\*a for which the spatially and temporally averaged simulated snowfall amounts for elevations below 2750 m a.s.l. match the respective*
*observation-based reference."*

*With this temperature correction you basically report the RCM snowfall to your reference. So, also in this case, before applying this procedure you*
*need an evaluation of the error on your snowfall reference. Moreover, can you give details on how you calculate spatial and temporal averages,*
*i.e. which domain/ time range?*

**Response and changes to manuscript** Concerning the evaluation of the snowfall reference please see our
replies above. Regarding the domain and time range, this information was actually provided two sentences
afterwards (Swiss domain and September to May). In the revised manuscript we moved this information to the
sentence you are referring to.

**Comment** *P9 L327: "Note that, as the underlying high-resolution data sets are available over Switzerland only, the calibration of the bias*
*correction methodology is correspondingly restricted, but the correction is then applied to the whole Alpine domain." HISTALP dataset provides*
*gridded snowfall monthly fields for the Alpine region at about 10 km spatial resolution (Chimani et al 2011), it should be included in the analysis*
*and discussed in comparison to your reference/manual observations in Switzerland: Chimani, B., Böhm R., Matulla C., Ganekind M.: Development*
*of a longterm dataset of solid/liquid precipitation Adv.Sci.Res,6,39-43, 2011http://www.adv-scires. net/6/39/2011/asr-6-39-2011.html*

**Response and changes to manuscript** Please see our replies above. HISTALP is now used to approximately
validate both the 2 km and the 0.11° snowfall reference. The paper of Chimani et al. is cited now. Thanks for
pointing us to this dataset (of which we have not been aware)!

**Comment** *P10 L338-341: "EURO-CORDEX simulations ... are compared against observations derived from measured fresh snow sums from 29*
*Meteo Swiss stations with data available for at least 80% of the evaluation period. For this purpose a mean snow density of 100 kg/m3 for the*
*conversion from measured snow height to water equivalent is assumed."*
*As said before, I am surprised to see at this point of the paper that you have 29 fresh snowfall time series covering the 1970-2005 period. They*
*should be presented before (section 3.1) & exploited much more than you do. These manned observations are the ground truth and they should*
*be used to validate the snowfall gridded dataset that you derive from temperature and precipitation over Switzerland. Please provide a quality*
*control of the snowfall gridded dataset prior to use it*

**Response and changes to manuscript** Please see our replies above. The comparison against fresh snow sums
can only be of approximate nature and is subject to large uncertainties. See the additional text in Chapter 3.1.
We'd refrain from considering these data as the "ground truth". These data are now introduced in Chapter 2.1
along with gridded observational data. They are also employed to approximately validate the 2 km and 0.11°
snowfall reference grid.

**Comment** *P10 L345-347: "The positive bias at high elevations might arise from the fact that the very few observations were made at a specific*
*location while simulated grid point values of the corresponding elevation interval might be located in different areas of Switzerland." Here you*
*consider all Switzerland and you really mix very different areas far away one from another. Please discuss the case when only the gridpoints*
*containing stations are considered, i.e. showing the spread of the models around the observed time series (i.e. in plots for the three - low, middle*
*and high - elevation ranges?*

**Response and changes to manuscript** Please see our replies above and Figure R1. In our opinion such a
comparison is subject to large uncertainties, mainly due to the mismatch of spatial representativeness between an
individual grid cell and an individual site. By design, such an evaluation can only be of approximate nature. See
also the additional text in Chapter 3.1 on the limitations of such a comparison. In our opinion, averaging over
elevation intervals and showing the respective spread (which covers or does not cover the site-scale
observations) is clearly the safer option.

**Comment** *P14 L487: "In between is a transition zone with rather strong changes with elevation". Can you explain why?*

**Response and changes to manuscript** This transition approximately corresponds to the mean elevation of the
SEP-MAY zero-degree line in today's climate. We added this information to the manuscript plus two additional
references. Elevations close to this line seem to be especially sensitive, which is in line with previous works
addressing future snow cover changes.

**Comment** *P14 L494-6: Could it also be residual biases along the snowfall line?*

**Response and changes to manuscript** We do not think so, as further analyses (see above; comparison of the
three approaches) indicate a robust relative climate change signal also at low elevations, no matter if raw (and
hence biased), separated or separated + bias-adjusted snowfall is analyzed.

**Comment** *P16 L587-8: Given more precipitation at high elevation & temperatures more favorable to heavy snowfalls, why does the snowfall*
*frequency decrease?*

**Response and changes to manuscript** This is explained in the brackets at the end of this sentence: The light
grey range (which is now explicitly mentioned in the caption of Figure 11) represents the temperature interval
below 2°C, i.e. approximately the interval where snowfall occurs at all (neglecting subgrid-scale effects and
assuming a binary threshold). Due to the general shift of the temperature distribution to the right (to higher
temperatures) the fraction underneath the red curve (scenario period) that falls into this interval is much smaller
than the one underneath the blue curve (control climate). This is equivalent to a decrease of the total snowfall
frequency, despite potential precipitation increases and higher mean snowfall intensities.

**Further technical corrections**

**Response and changes to manuscript** Thank you very much for these additional comments and ideas. All
suggested further technical corrections were implemented with the exception of: (1) P3 L79 "low" to "lower" (which
is not meaningful in our opinion), (2) suggested changes to Figure 6 (an important point here is the differing
spatial distribution of mean snowfall, this would not be apparent in figures of anomaly wrt. the reference), (3) P8
L280 (the respective text section refers to the Richards method, here the spread is indeed +/- 0.1 at maximum)
and (4) Figure 11 (this figure is schematic only, low and high elevations are not defined in detail).

**Further changes to the manuscript**

**Chapter 2.1 "Observational data"** For reasons of completeness we additionally included the information that the temperature and precipitation grids employed are slightly shifted with respect to their reference time interval (midnight UTC - midnight UTC for temperature, 06 UTC - 06 UTC for precipitation).

**Chapter 2.2 "Climate model data"** In the last paragraph we erroneously spoke of *six* RCMs considered. We corrected this to *seven* RCMs.

**Chapter 3.1** To better account for uncertainties in this simplified evaluation we now additionally cite the work of Grünewald and Lehning (2015) that highlights the danger of non-representativity of single-site snow depth observations in Alpine terrain.

**Figure 3** In the left panel two of the 29 stations employed (WFJ and MVE) were plotted at a wrong elevation in the original version. For both stations the correct elevation differs by about 100 m from the previously used elevation. The figure has been corrected accordingly, in addition to the modifications to this figure mentioned above. The conclusions of the analysis do not change.

**Overall manuscript** Several minor spelling and wording mistakes as well as an inconsistent use of past and present tense were corrected.

[revised manuscript text omitted]

---

## Author Comment (AC3) · 30 May 2017

**Future snowfall in the Alps: Projections based on the EURO-CORDEX regional climate models**

Prisco Frei, Sven Kotlarski, Mark A. Liniger, Christoph Schär

**- Response to Referees –**

**General**

We thank the three referees for their careful revision of the manuscript and for their constructive comments. Please find below our replies to all major comments and our suggestions on how to address these issues in a revised manuscript. We hope that we satisfactorily addressed all referee comments and that the proposed changes are considered as being appropriate. In case that not, we'd be looking forward to discuss individual remaining issues in more detail.

As several referee comments addressed the RCM evaluation and the evaluation of the 2 km snowfall reference itself, we'd like to put in front the following two statements on the scope of the paper:

(1) Our work is primarily concerned with the analysis of future snowfall projections. However, a basic notion on the quality of raw RCM snowfall and, hence, the general ability of RCMs to represent our variable of interest is required for such an exercise in our opinion. In the manuscript this is accomplished by comparing RCM raw snowfall against site-scale measurements obtained from new snow sums (Figure 3). Such a comparison is subject to considerable uncertainties, mostly originating from the scale gap between RCM grid cells and site-scale observations and from representativity issues of observed snow cover. Due to a missing high-quality observational reference at the scale of the RCM resolution (in our opinion also the HISTALP dataset has its shortcomings; see below) we refrain from evaluating RCM snowfall in more detail and, at least when interpreting raw snowfall change signals, implicitly assume stationary model biases. As a consequence, the projection aspect of the current work is much larger than the evaluation aspect, and we tried to better clarify this issue by modifying the text in Chapters 1 and 3.1. Furthermore, we adjusted the title of the manuscript accordingly and removed the term "Evaluation".

(2) Relating to the previous issue but especially to the validation of the snowfall reference against which the RCM-derived snowfall is adjusted: As it was already mentioned in the introduction of the original manuscript, we do not claim to present an ultimate solution for bias-adjusting RCM-based snowfall but employ a spatially and temporally aggregated adjustment procedure that does nevertheless separately account for temperature and precipitation biases. Aspects of the snowfall climate that are not corrected for, such as details of the spatial snowfall pattern, are described in Sections 2.6 and 3.3. The simplifications also include the fact that we basically accept a non-perfect observation-based reference. A well-validated and appropriate reference does not exist in our opinion (see also above). The very core of our work is the analysis of projected future snowfall changes and the comparison of three different ways to produce such estimates: (1) Raw RCM snowfall, (2) RCM snowfall as separated from simulated temperature and precipitation, and (3) RCM snowfall as separated from simulated temperature and precipitation and additionally bias-adjusted. The latter version is the basic dataset for the climate scenario analysis as it can be constructed for all participating RCMs (raw snowfall is not available for all of them) and as it is, in principle, able to account for temperature-dependent and hence non-stationary snowfall biases. However, Chapter 5.3 shows that relative change estimates largely agree among all three datasets and are robust. From that point of view, the influence of remaining inaccuracies in the bias-adjusted snowfall projections due to inaccuracies of the reference is presumably small. In the revised version we now try to better clarify these issues by modifying Chapters 1, 2.5, 2.6 and 5.3.

**Response to Referee #3**

**Comment** *Half-way the introduction (lines 75-80) the authors write " Within the last few years …" followed by "Most of these analyses are based on GCM output or older generations of RCM ensembles at comparatively low spatial resolution : : :". This may indeed be the case for most of the studies cited in the sentence before, but not for all of them, and the authors should specify explicitly which of them is the exception, and how that study. compares with their work. E.g. Piazza et al. use, amongst other models, a number of RCMs operated at 12 km resolution, the study by de Vries et al. (2014) is based on a 8-member ensemble of EC-EARTH-RACMO simulations at 12km resolution (historical and rcp8.5) configured on a smaller domain, but in principle quite comparable to the simulations used in this paper.*

**Response and changes to manuscript** You are perfectly right, thanks for pointing this out. We added further information on the existing RCM-based studies in this section. Furthermore, reference to the mentioned works had already been given in the conclusions of the original manuscript.

**Comment** *Instead of using "bias correction" I would strongly recommend to use the phrase "bias adjustment" as was also the adopted terminology by the EURO-CORDEX community. The word "correction" suggests there is a well-established methodology including a ground truth observed*

*state which we all agree on. This is obviously not the case. It is therefore much better to use the word "adjustment" which automatically triggers*
*the questions "how" and "to what" or "in which context" as it should be.*

**Response and changes to manuscript** We are aware of the current discussion on this issue and basically
agree with the referee's suggestion. There are pros and cons to it, however. The term "bias correction" is currently
used and better understood by a wider community and is also better reflected by the available literature. But
exactly due to the points mentioned we agree that "bias adjustment" is the more suitable term. We therefore
changed "bias correction" ("bias-corrected") to "bias adjustment" ("bias-adjusted") throughout the entire
manuscript. This also involves modifications to the legend of Figure 13. Accordingly, we also changed "RCM$_{sep+bc}$"
("RCM$_{sep+nbc}$") to "RCM$_{sep+ba}$" ("RCM$_{sep+nba}$"). In Chapter 2.6 we include the following sentence outlining the
reasons behind the choice of the term: "Note that we deliberately employ the term bias adjustment as opposed to
bias correction to make clear that only certain aspects of the snowfall climate are adjusted and that the resulting
dataset might be subject to remaining inaccuracies."

**Comment** *In section 2.2 it is mentioned that all GCM-driven EUR-11 simulations for which control, RCP4.5 and RCP8.5 runs are currently*
*available have been included in the study. This can obviously not be a correct state of affairs. Currently means "at the moment" and since the*
*number of simulations published in the ESGF-archive is still growing, there will be a moment that the statement is no longer true. In fact, already in*
*October 2016 there were 16 simulations that met the criteria set by the authors. In addition to the their selection there were results published from*
*HadGEM2-RACMO and MPI-ESM (r2i1p1)-REMO. In April 2016, none of the two MPI-ESM-REMO simulations were available, but the HadGEM2-*
*RACMO simulation was, albeit based on version v1 which was replaced by version v2 in August 2016. So, a) you need to specify currently, and b),*
*either include the simulations that were in the archive but not in your selection, or convincingly explain why some simulations were not selected.*

**Response and changes to manuscript** You are completely right, thanks for pointing this out. We're now providing the
date on which the ESGF database was accessed for the purpose of this paper. And we also provide the reasons for not
including two of the available experiments. HadGEM2-RACMO was disregarded due to serious snow accumulation issues over
the Alps with obvious feedbacks on temperature (which is used in the snowfall separation process). MPIM-REMO realization 2
was disregarded as our purpose was to assess model uncertainty by employing an ensemble analysis and not internal climate
variability (realization 1 is contained in our ensemble, see also footnote of Table 1 in the original manuscript).

**Comment** *Following the previous point, it is important mentioning that three different realizations of ICHEC-EC-EARTH are used to force four*
*different RCMs: r12i1p1 is used to force CCLM and RCA, r3i1p1 is used for HIRHAM, and r1i1p1 for RACMO. Different realizations of a GCM can*
*show distinct behavior owing to long-term large-scale natural variability implying that differences between the EC-EARTH forced RCM simulations*
*are not only due to differences in the RCMs. This would only hold for the CCLM end RCA simulations forced with r12i1p1. Please, mention this*
*aspect when introducing the 14 GCM-RCM chains.*

**Response and changes to manuscript** Thanks a lot, this fact is now mentioned in the Caption of Table 1. It is,
however, of minor importance for the present study, as for instance the influence of the driving GCM is not
analyzed in detail but only mentioned at one point in the manuscript.

**Comment** *In section 2.5 the methodology to separate snowfall from total precipitation is discussed (Richards method). In the final paragraph a*
*parametric formulation $f_{s,Ri}$ is introduced (Eqs 4-7) to express the snowfall fraction in terms of coarse-grid temperature $T_k$, the topographic*
*standard deviation h of the designated grid cell, and a number of constants (E,F,G,H) which are determined through an empirical fitting procedure.*
*The function $f_{s,Ri}$ is meant to be used to separate snowfall in the RCMs as well (line 282-283). Since the subgrid-scale orographic variance*
*parameter of the model orography is not known (at least this parameter is not in the ESGF-archive) I presume the authors have used h from the*
*observational dataset. Somehow the observational h and model height should match. This is not guaranteed a priori. The authors already mention*
*that orography fields from different RCMs can be quite different from each other, and from the observations (line 157-159). The authors should*
*explain how they deal with such mismatches.*

**Response and changes to manuscript** The origin of the "observed" high resolution topography (GTOPO30) is
now properly referenced. This information was indeed missing so far. Concerning a potential mismatch between
mean grid cell topography of the RCM and mean grid cell topography as obtained from GTOPO30 we do not
believe that this leads to any problem. The parameterizations in the Richard method rely on topographic variance
at a subgrid level only, not on mean topography. We assume that this variance is properly represented by
GTOPO30. In any case, the relation is later on fitted again (see Figure S1). Whether the models internally work
with a similar topographic variance or not is not relevant here in our opinion. Nevertheless we now explicitly
mention this apparent and potential mismatch of mean orographies in Section 2.5.

**Comment** *In section 2.6 the bias adjustment approach is discussed. I was surprised to see that after the detailed treatment of snowfall separation,*
*the adjustment of temperature has been dealt with so crudely. While according to Fig S4 the temperature bias considerably depends on elevation*
*the authors have chosen the shifted fractionation temperature to be independent of elevation. According to Fig 4 the adjusted fractionation*
*temperature and the temperature bias (one point per GCM-RCM realization) show considerable scatter around a linear relation, but it is not at all*
*clear and also not explained what causes this scatter. It might be due to bias depending on elevation, but also to month of the year and/or region.*
*Or is there something else? The authors should discuss their treatment of shifting the fractionation temperature in more detail, it is particularly*
*relevant because the snow fractionation temperature appears to be a crucial, and probably also a sensitive, parameter in the analysis.*

**Response and changes to manuscript** Thanks very much for pointing out this issue. We agree that our bias
correction is approximate only. For instance, it only targets spatially and temporally averaged mean snowfall but
not the spatial pattern (see Figure 6). These limitations are actually mentioned. More sophisticated methods can
be thought of, but bear the danger of overparameterisation and often require a more accurate observational
reference. The bias adjustment of snowfall in the frame of the present paper is only one of three postprocessing
methods (in addition to raw = no postprocessing and separation only but no bias adjustment). See also our replies
in the very beginning of this document. In the end, we find that relative change signals of snowfall indices closely
agree no matter what the postprocessing is. Concerning the adjustment of the fractionation temperature: the bulk
adjustment for the entire domain is mainly based on our target (domain mean snowfall). Adjusting separately for
individual elevation intervals might be a better account of elevation-dependent temperature biases but would probably over-interpret the uncertain snowfall reference in terms of its elevation dependency. This is mentioned now in the revised version of the manuscript (Section 2.6). Regarding the relation between mean temperature bias and adjusted fractionation temperature: There are several potential reasons for deviations from a linear relation. In addition to the ones mentioned in your comments, also differences in daily temperature variability or in the bivariate temperature-precipitation distribution can be thought of. In the revised manuscript we now mention potential reasons in more detail.

**Comment** *Line 565-566: The sentence "Previous studies : : : with this theory (e.g. Allen and Ingram, 2002; Ban et al. 2015)" is completely out of context. I strongly suggest to omit this sentence and the corresponding references. The focus of the Ban et al. paper is on summertime convectively driven sub-daily (hourly) precipitation extremes and its relation with temperature. This is miles away from the Sq99 and S1d snowfall parameters used in this paper to indicate heavy (but not extreme) snowfall at the daily scale outside the summer season.*

**Response and changes to manuscript** It is certainly true that we neither deal with sub-daily extremes nor with the very tail of the daily snowfall distribution. However, we still believe that this information is relevant and not out of context as is presents general evidence for applicability of the C-C-relation concerning precipitation extremes. We'd therefore like to refrain from removing this sentence, but tried to better clarify the fact that different variables are addressed in the cited works.

**Further minor comments**

**Response and changes to manuscript** Thank you very much for these additional comments and ideas! All suggested further minor corrections were implemented with the exception of comment 28 (Sorry, this one is not clear to us) and comment 29 (If left up to us, we'd prefer to stick to the current color scheme which is intuitive in our opinion and still allows to grasp the important characteristics of the spatial distribution).

**Further changes to the manuscript**

**Chapter 2.1 "Observational data"** For reasons of completeness we additionally included the information that the temperature and precipitation grids employed are slightly shifted with respect to their reference time interval (midnight UTC - midnight UTC for temperature, 06 UTC - 06 UTC for precipitation).

**Chapter 2.2 "Climate model data"** In the last paragraph we erroneously spoke of *six* RCMs considered. We corrected this to *seven* RCMs.

**Chapter 3.1** To better account for uncertainties in this simplified evaluation we now additionally cite the work of Grünewald and Lehning (2015) that highlights the danger of non-representativity of single-site snow depth observations in Alpine terrain.

**Figure 3** In the left panel two of the 29 stations employed (WFJ and MVE) were plotted at a wrong elevation in the original version. For both stations the correct elevation differs by about 100 m from the previously used elevation. The figure has been corrected accordingly, in addition to the modifications to this figure mentioned above. The conclusions of the analysis do not change.

**Overall manuscript** Several minor spelling and wording mistakes as well as an inconsistent use of past and present tense were corrected.

[revised manuscript text omitted]
 adjusted simulated snowfall ($RCM_{sep+nbae}$), and the separated and bias-corrected adjusted simulated snowfall ($RCM_{sep+bae}$). In the first case the snowfall separation of raw precipitation is performed with $T^*$=2°C, while in the second case precipitation is corrected adjusted and the separation is performed with a bias-adjusted temperature $T^*_a$. The first column represents the mean September to May statistics, while columns 2-4 depict the seasonal cycle at monthly resolution for three distinct elevation intervals.

The analysis of $S_{mean}$ confirms that $RCM_{sep+bae}$ is able to reproduce the observation-based reference in the domain mean as well as in most individual elevation intervals. The domain-mean agreement is a direct consequence of the design of the bias correction adjustment procedure (see above). $RCM_{sep+nbae}$, on the other hand, consistently overestimates $S_{mean}$ by up to a factor of 2.5 as a consequence of positive precipitation and negative temperature biases (cf. Fig. 4). Also the seasonal cycle of $S_{mean}$ for $RCM_{sep+bae}$ yields a satisfying performance across all three elevation intervals, while $RCM_{sep+nbae}$ tends to produce too much snowfall over all months and reveals an increasing model spread with elevation.

For the full domain and elevations around 1000 m, the observation-based reference indicates a mean $S_{freq}$ of 20% between September and May. Up to 1000 m a.s.l. $RCM_{sep+bae}$ reflects the increase of this index with elevation adequately. However, towards higher elevations the approximately constant $S_{freq}$ of 30% in the reference is not captured by the simulation-derived snowfall. Notably during wintertime, both RCM$_{sep+b\sim\text{a}e}$ and RCM$_{sep+nb\sim\text{a}e}$ produce too many snowfall days, i.e., overestimate snowfall frequency. This feature is related to the fact that climate models typically tend to overestimate the wet day frequency over the Alps especially in wintertime (Rajczak et al., 2013) and that the bias  adjustment procedure employed does not explicitly correct for potential biases in precipitation frequency. Due to the link between mean snowfall on one side and snowfall frequency and mean intensity on the other side, opposite results are obtained for the mean snowfall intensity $S_{int}$. RCM$_{sep+b\sim\text{a}e}$ largely underestimates mean intensities during snowfall days while RCM$_{sep+nb\sim\text{a}e}$ typically better reflects the reference. Nevertheless, deviations during winter months at mid-elevations are not negligible. Mean September-May $S_{frac}$ in the reference exponentially increases with elevation. This behaviour is reproduced by both RCM$_{sep+b\sim\text{a}e}$ and RCM$_{sep+nb\sim\text{a}e}$. Notwithstanding, RCM$_{sep+b\sim\text{a}e}$ results are more accurate compared to RCM$_{sep+nb\sim\text{a}e}$, which turns out to be biased towards too large snowfall fractions.

For the two heavy snowfall indices $S_{q99}$ and $S_{1d}$, RCM$_{sep+nb\sim\text{a}e}$ appears to typically match the reference better than RCM$_{sep+b\sim\text{a}e}$. Especially at high elevations, RCM$_{sep+b\sim\text{a}e}$ 
[revised manuscript text omitted]

---

## Referee Report (RR2)

MS No.: tc-2017-7, revised-2 manuscript
Review, 2 October 2017

Title: Snowfall in the Alps: Evaluation and projections based on the EURO-CORDEX regional climate models

Authors: Prisco Frei, Sven Kotlarski, Mark A. Liniger, Christoph Schär

Recommendation: [Revision]

GENERAL COMMENTS:

To start with, the authors have adequately dealt with the $1^{st}$ and $3^{rd}$ major point I raised in my review of the first revised manuscript.

However, their response to the $2^{nd}$ major point came as a big surprise to me, even more so because the editor had pointed out an elegant way to solve the issue of not presenting the results from the HadGEM2-ES forced RACMO simulation. Instead the authors have chosen to take out the ECEARTH-RACMO and the IPSL-WRF simulation based on a new line of argumentation.

The authors now state that snow depth issues themselves are not the real problem in their study but rather the adverse feedback from snow depth issues on future temperature trends, which cannot be adequately dealt with by their bias-adjustment method. That may be so, yet I really wonder why they haven't considered this aspect right from the beginning. I also want to remind them that in the first revised manuscript they only used the snow depth issue as an argument to exclude a model chain without mentioning the role of temperature: "The HADGEM2-KNMI experiments were excluded due to serious snow accumulation issues over the European Alps." (part of footnote in section 2.2). (Snow accumulation issues is unfortunate phrasing, snow depth issues is much more adequate in this respect.)

According to Fig R1 in the rebuttal the RACMO- and WRF-simulations in particular suffer from adverse temperature effects owing to problems in snow depth. I won't dispute that there is a problem, but as I argued in my previous review this is primarily apparent in summer. The authors agree on that, but because problems in the transition seasons cannot be ruled out they have decided to take out both model chains altogether.

However, what the authors forget to mention in their rebuttal –and that is really unfortunate - is that their Fig R1 shows the temperature change signal of four models for the *summer* season, which is precisely the season they had left out from their analysis. What they should have done instead is displaying the temperature change signal for the rest of the year (September-May).

To underline my point I have generated two figures showing the temperature change signal for all models involved (using the ordering in their original manuscript and $1^{st}$ revision, supplied with HadGEM2-RACMO in the $15^{th}$ panel) for the summer season (Figure A) and the rest of the year (Figure B). To facilitate the comparison, I have tried to match the spatial extract and color scheme used in the Fig. R1 as much as possible. I've added distinct colors for values outside their color bar below +1K and above +7K.

The JJA temperature change patterns in Figure A for the model-chains ECEARTH-RACMO, HADGEM2-RACMO, ECEARTH-HIRHAM, IPSL-WRF, are very similar to the patterns displayed in the author's figure Fig. R1, though there seems a discrepancy for ECEARTH-

HIRHAM, perhaps related to the fact that there are quite a few points for this model chain with temperature changes below +1K outside the range displayed in Fig R1. Nonetheless , the resemblance for the other three model chains is striking and this convincingly shows that Fig. R1 indeed corresponds to the summer season. Note also that the model chains including RCA4 have issues as well in JJA temperature change signal with mostly (very) high values over the Alps alternated with some very low values, probably for grid cells at the highest elevations.

Yet, the results in Fig. A or Fig. R1are simply not relevant for this manuscript, because the authors have, for good reasons, confined their analysis to the September-May period of the year. The temperature change signals of the 15 model chains, displayed in Figure B, show a mixture of very smooth patterns (in particular the chains involving CCLM) to somewhat more varying patterns for the other model chains, most of the variation clearly induced by variation in altitude. However, none of the patterns shows disturbingly large and/or spurious variations.

From Figure B, I conclude that the temperature change signals derived for the September-May period of the year offer no ground to exclude any of the model chains from the analysis. If, based on this figure you decide to take out ECEARTH-KNMI from the analysis, I would argue that you should remove for the very same reason the model chains involving RCA4, REMO2009, ALADIN, and probably HIRHAM5 as well.

I strongly urge the authors to stick to the original selection of model chains, including ECEARTH-RACMO and IPSL-WRF, and add HADGEM-KNMI to the selection, because there is no (scientific) reason to leave it out. I also checked that IPSL-WRF output has not been withdrawn from EGSF, so I don't see a good reason to take out that model chain after you had submitted the initial manuscript.
For future reference it would be very useful to add a paragraph to the manuscript pointing out that some individual model chains suffer from snow depth issues which have detrimental impacts on near-surface temperatures in the summer season, but that such behaviour is hardly apparent in the rest of the year, and hence does not adversely affect your analysis. As the editor already pointed out the previous time you can use the supplementary information to elaborate on that.

Finally, the role of the upcoming CH2018 Swiss scenarios is interesting but not really decisive for the choices made in the manuscript. First, CH2018 has a much broader context than the study this manuscript reports on. Also CH2018 serves another purpose and audience than the scientific journal in which this manuscript is meant to be published. Lastly, and relevant here, the original manuscript was submitted in the beginning of 2017, prior to publication of the SH2018 model set documentation. Moreover, in the original manuscript there was no mentioning whatsoever of CH2018. (Why not?) So, in my opinion the decisions made in preparation of CH2018 do not provide justifiable reasons to change the selection of model chains during an ongoing review and revision procedure.

[Figure]

Figure A: Temperature change signal for the summer season (JJA) over the Alpine region (2070-2099 wrt. 1981-2010) for all EUR-11 model chains considered in the original manuscript (and 1[st] revision) (Table 1) supplied with HADGEM2-RACMO, assuming RCP8.5 from 2006 onwards.

[Figure]

Figure B: Like Figure A, but for the September-May period of the year (SON+DJF+MAM)

---

## Author Response (AR2)

**Future snowfall in the Alps: Projections based on the EURO-CORDEX regional climate models**

Prisco Frei, Sven Kotlarski, Mark A. Liniger, Christoph Schär

**- Second response to Referees –**

**General**

We thank both referees for another thorough check of the revised manuscript. We're happy about the overall positive feedback, but also sorry about the fact the we could not address all concerns of referee #3 satisfactorily so far. As a consequence we have further revised the manuscript, please see the more detailed replies below. The most important change to the paper is a revised set of climate model simulations that are considered. All suggested further minor changes have been incorporated. We hope that the updated version of the paper raises no more concerns.

With kind regards, Sven Kotlarski (on behalf of all co-authors)

**Response to Reviewer 2**

**Comment** *Dear Authors, the revised manuscript has been improved a lot, all the major points have been adequately addressed in the reply to the Reviewers and eventually integrated in the main text. After the correction of the few typos listed below, I think this version of the manuscript is suitable for publication.*

*Congratulations for this nice paper!*

*Kind regards*

**Response and changes to manuscript** Thank you very much for the positive feedback. All suggested changes to the manuscript except have been incorporated.

**Comment** *Line 96: throughout the annual cycle -> throughout the year".*

**Response and changes to manuscript** Corrected.

**Comment** *Line 105: "on an RCM grid" -> on "the" RCM grid.*

**Response and changes to manuscript** Corrected.

**Comment** *Line 129: "the very focus" -> "the focus"*

**Response and changes to manuscript** Corrected.

**Comment** *Line 130: "on the snow projection aspect" -> "on snow projections"*

**Response and changes to manuscript** Corrected.

**Comment** *Line 138: "To estimate observation-based snowfall" sounds like an oxymoron, I suggest: "To estimate the reference fine-scale snowfall"*

**Response and changes to manuscript** Corrected.

**Comment** *Line 230: "not all RCM simulations available through EURO-CORDEX provide raw snowfall" -> "not all EUROCORDEX RCMs provide raw snowfall"*

**Response and changes to manuscript** Corrected.

**Comment** *Page 12: missing reference to Figure S5.*

**Response and changes to manuscript** Sorry, this was indeed a mistake. The figure should have been referenced in Section 3.2. We have now added the reference (note that the Figure is now called S6 instead of S5).

**Comment** *Line 452: "partly explain the partly substantial overestimation"*

**Response and changes to manuscript** Corrected.

**Comment** *Line 713: "In" -> in*

**Response and changes to manuscript** Corrected.

**Comment** *Line 731: "the an"*

**Response and changes to manuscript** Corrected.

**Comment** *Line 735: "on an" -> "on the"*

**Response and changes to manuscript** Corrected.

**Comment** *Line: 783: "This effect … may be strong under conditions (depending upon location and season) where the current*
*climate is well below freezing. Such conditions may experience a shift towards a temperature range more favourable to*
*snowfall". Maybe the word "conditions" is not the best choice here, can you please rephrase?*

**Response and changes to manuscript** You are perfectly right. We rephrased the second sentence.

**Comment** *Figure 2: the caption is unclear, you could rephrase "Ratio of a) mean snowfall Smean, b) heavy snowfall Sq99*
*obtained with the Binary and the Richardson methods to the corresponding values obtained with the Subgrid method …"*

**Response and changes to manuscript** We did not change this specific sentence, but clarified the previous
sentence (the meaning of the ratios is explained in there).

**Response to Reviewer 3**

**Comment** *In my review of the original manuscript a couple of months ago I raised a couple of major points which in my opinion*
*needed attention. Having read the revised manuscript and the accompanying rebuttal letter I notice that the authors have done*
*a considerable effort to address all issues, also those from the other reviewers. Overall, the revisions have resulted in an*
*improved manuscript. Unfortunately, not all comments have been adequately addressed, therefore some further attention is*
*needed before the manuscript is suitable for publication.*

**Response and changes to manuscript** Thanks very much for this feedback on our revisions. We're sorry that
individual comments were not properly addressed in the reviewer's opinion. In the revised version we tried to
further improve the manuscript in this respect and tried to incorporate all suggested changes. We hope and
believe that the newly revised version is acceptable now.

**Comment** *1. (also point 1 in first review) The sentence added by the authors to point out that some works also utilized output*
*from high resolution (12km) RCM output "It thereby complements …(partly originating from EURO-CORDEX) … but with a*
*reduced ensemble size and/or not specifically targeting the entire Alpine region" I don't understand both phrases.*

*Firstly, EURO-CORDEX has a marginal role in both papers, and certainly nothing of the material and methodology originates*
*from it. Just leave it out.*

*Secondly, these are both quite different studies, the Piazza et al. paper studies a variety of models in their reponse for a region*
*in the French Alps., while the de Vries et al. paper utilizes a single-model 8-member ensemble considering a region including*
*the entire Alpine region. I would advise to just explicitly phrase in what these papers primarily differ from your paper (i.e. region*
*of analysis in Piazza et al. paper, and nature of model ensemble studied in de Vries et al.) and avoid using "and/or"*
*constructions*

**Response and changes to manuscript** We're sorry that our original corrections are not appropriate in the
reviewer's opinion. We agree with the further points raised and modified the text according to the new
suggestions.

**Comment** *2. (point 3 first review). I do not disagree that HadGEM2-RACMO has an issue with snow accumulation over the*
*Alps, but this is rather the manifestation from a missing process (no glacier model) in combination with not setting an artificial*
*limit on maximum snow depth (as is done in some of the other RCMs), than too much snow fall. Very large numbers in snow*
*depth (or snow mass if available) are also seen in the model combinations EC-EARTH-HIRHAM, EC-EARTH-RACMO, and*
*IPSL-WRF (in the first two even larger than in HadGEM2-RACMO), yet these combinations have been retained.*

*The authors furthermore mention in their rebuttal that the "obvious feedbacks on temperature" led them to disregard the*
*HadGEM2-RACMO simulation. ". I would expect this to hold primarily for the summer season, and far less relevant in the winter*
*season when sub-zero temperatures still dominate at the higher elevations where the anomalously high snow accumulation is*
*found.*

*But even when it also affects the September-May season studied by the authors I don't' see why the other three model*
*combinations, mentioned above, don't suffer from this problem. After all, the sensitivity to temperature bias is to a large extent*
*taken care of by the (bias-adjustment) shift in snow fractionation temperature T\* to Ta\*.*

*In my opinion the reason to disregard HadGEM2-RACMO is formulated rather ad hoc and could have been equally well been*
*applied to disregard other model combinations which have now been retained. That leaves three options: 1) include HadGEM2-*
*RACMO in the analysis, 2) disregard the other three combinations as well, or 3) provide a plausible explanation why this specific*
*model combination was disregarded and other combinations not.*

**Response and changes to manuscript** We are sorry that our original revisions to the paper were not considered
as appropriate. We partly agree to the remaining comments of the reviewer. However, as we tried to describe in
our previous replies, the snow accumulation in individual models itself is not problematic for our study as we're
not employing simulated snow depth or simulated snow water equivalent. The problem here is the apparent
feedback on the 2m temperature with adverse effects on the future temperature trend at individual grid cells. This
problem cannot be overcome by our bias correction which is trained in the reference period and which does not
account for non-stationary model biases. Hence, a widespread adverse influence on temperature trends in the
analysis domain is an obvious reason to exclude the affected model chains from the analysis. The reviewer is
right that this problem is most apparent in summer, but we cannot rule out effects in the transition seasons. The
reviewer is also right in mentioning that further model chains are affected. This particularly concerns ECEARTH-
RACMO (about 100 grid cells in the Alpine affected by snow accumulation), we are thankful for this comment. The
problem is far less pronounced, however, for ECEARTH-HIRHAM (about 20 grid cells affected) and for IPSL-
WRF (same order). Please see Figure R1 below for the spatial pattern of temperature changes by the end of the
century for RCP8.5 and the four model chains under discussion. A widespread feedback of snow accumulation on
2 temperature trends is obtained for both RACMO experiments. We therefore took these simulations out of our
revised analysis. Following a further shortcoming of the IPSL-WRF experiment that was discovered in meantime
(unphysical temperature change patterns) we also removed this model chain from our analysis, resulting in a total
of 12 model chains analyzed (instead of 14). As a consequence, all results figures of the manuscript were
adjusted, but the basic patterns do not change and the conclusions of the manuscript are not modified. The
selection of experiments to be removed is now fully consistent with the upcoming CH2018 Swiss climate
scenarios, it is documented on the CH2018 website www.ch2018.ch and in a specific model set documentation
available from http://www.ch2018.ch/wp-content/uploads/2017/07/CH2018_model_ensemble.pdf. The latter also
provides more details on the reasons for removing individual experiments. The text Section 2.2 of the manuscript
has been revised accordingly, as well as Table 1. A reference to the CH2018 website is provided.

[Figure]

**Fig. R 1** Temperature change signal over the Alpine region (2070-2099 wrt. 1981-2010) for four specific EUR-11 model chains (naming:
GCM-RCM) and for RCP8.5.

**Comment** *3. (point 7 first review).The way the authors have addressed this point only makes things worse. I repeat that the line*
*"Previous studies … with this theory (e.g. Allen and Ingram, 2002; Ban et al. 2015)" is out of context, for reasons the authors*
*agree on but refuse to act upon. What the authors do is adding a phrase which underlines that the material in their paper (daily*

*scales, no convection, cold season) is incompatible with the conditions studied in the Ban et al paper (sub-daily or hourly scales,*
*convection, warm season), yet they state that there finding is consistent with the Ban et al..*

*An additional reason why I am emphasizing this point is that the literature on the projected behavior of hourly precipitation*
*extremes under climate change is quite controversial, and far from settled at the moment (see also literature cited in Ban et al.*
*2015). That aspect is entirely ignored in the current formulation, which is unacceptable.*

*I therefore stick to my point made in the first review that this sentence including the corresponding references must be taken out.*

**Response and changes to manuscript** We apologize, it appears this is a misunderstanding. It is correct that the
main emphasis of Ban et al. (2015) is on hourly precipitation scaling (and this is not addressed in the current
study). However, the paper also considered the scaling of daily precipitation events. It has not been the intent of
our revisions to focus on hourly time scales. We have clarified the statements correspondingly. To avoid
confusion, we now reference a recently accepted publication (Rajczak et al. 2017) that entirely addresses daily
time scales, instead of Ban et al. (2015).

**Comment** *(point 28 in the first review) I meant to add an ancillary line showing the frequency of occurrence of the elevation*
*intervals across the region of interest (either Switzerland or the Alpine region or both). Evidently the higher elevations are (far)*
*less abundant than the lower elevations – there will be only very few grid cells with elevation beyond 3000 m a.s.l. , making the*
*analysis for high elevation less robust. It would help to give an impression of the pdf of the elevations, you could use the upper*
*horizontal axis to label the percentage. The attached png-file shows what I meant to say (only intervals above 250m are*
*counted), the authors could plug in such line in Fig 3a, or, if they prefer in Fig 2, or Fig 4a, that is for them to choose.*

**Response and changes to manuscript** Thanks very much for these further explanations, and sorry that the
comment of the original review was not clear to us. We agree that this information would be helpful. However, we
would prefer to introduce a separate figure on the area-elevation distribution rather than overloading existing
figures with an additional line and an additional axis. We therefore added an additional figure (S2) to the
supplementary material that shows the area-elevation distribution of the entire Alpine analysis domain based on
the aggregated GTOPO30 elevation model. The figure is now referenced in Section 3.1 of the manuscript.

**Comment** *(major point 4 first review) Indicate in Table 1 which EC-EARTH realizations are used (like you do for MPI-ESM-*
*REMO) in the various EC-EARTH-RCM combinations: r1i1p1 for EC-EARTH-RACMO; r3i1p1 for EC-EARTH-HIRHAM; r12i1p1*
*for EC-EARTH-SMHI and EC-EARTH-CCLM.*

**Response and changes to manuscript** Done, thanks a lot. In addition, we provide the information on the
realization of the driving GCM now in the column "GCM" instead of the column "Acronym". This is more
appropriate in our opinion.

**Comment** *Line 267: "leads a"* ☐ l̃eads to a"
**Response and changes to manuscript** Corrected.

**Comment** *Line 300: rephrase "and as we condere σh as a" as "while σh is regarded as a …"*
**Response and changes to manuscript** Corrected.

**Comment** *Line 328: rephrase "under current and future … climate conditions" as "under changing … climate conditions*
**Response and changes to manuscript** Corrected.

**New manuscript version with marked-up changes:**

[revised manuscript text omitted]

**Comment [Sven11]:** Figure revised due to removal of two further model chains.

**Figure S3**: Ratios (RCM simulations divided by observational analysis) of total precipitation sums from September to May in 1971 - 2005 vs. elevation for the Swiss domain. The linear regression line, applied to the ratios for elevations between 250 m a.s.l. and 2750 m a.s.l., is represented by the red line.

[Figure]

**Comment [Sven12]:** Figure revised due to removal of two further model chains.

**Figure S4**: Total winter (SEP-MAY) precipitation bias (expressed as quotient between RCM simulations and
observations) in the EVAL period 1971-2005 for individual elevation intervals and for the full domain (lowermost
row). Left panel: Swiss domain only. Right panel: Entire Alpine analysis domain (cf. Fig. 1). Observational
reference: EOBS version 13.1 (Haylock et al., 2008) on 0.22° interpolated to the 0.11° RCM grid by nearest
neighbour interpolation.

[Figure]

**Comment [Sven13]:** Figure revised due to removal of two further model chains.

**Figure S5**: As Figure S4 but for the winter (SEP-MAY) temperature bias.

[Figure]

**Figure S6** Mean September-May snowfall sum [mm] in the period 1971-2005 as represented by the 2 km
snowfall reference (*Subgrid method*; upper left), the 12 km snowfall reference on the RCM grid (*Richards method*;
upper right) and the HISTALP dataset (Chimani et al., 2011; lower).

[Figure]

Comment [Sven14]: Figure revised due to removal of two further model chains.

**Figure S7** Spatial distribution of relative changes (SCEN period 2070-2099 with respect to CTRL period 1981-
2010) in mean September-May snowfall, $\delta S_{mean}$, for RCP4.5 and for the 12 snowfall separated + bias-adjusted
RCM simulations (RCM$_{sep+ba}$).

[Figure]

Heavy snowfall change ($\delta S_{q99}$) [%], RCP4.5

**Comment [Sven15]:** Figure revised due to removal of two further model chains.

**Figure S8** Spatial distribution of relative changes (SCEN period 2070-2099 with respect to CTRL period 1981-
2010) in heavy snowfall, $\delta S_{q99}$, for RCP4.5 and for the 12 snowfall separated + bias-adjusted RCM simulations
(RCM$_{sep+ba}$).

[Figure]

Heavy snowfall change ($\delta S_{q99}$) [%], RCP8.5

**Comment [Sven16]:** Figure revised due to removal of two further model chains.

**Figure S9** Spatial distribution of relative changes (SCEN period 2070-2099 with respect to CTRL period 1981-2010) in heavy snowfall, $\delta S_{q99}$, for RCP8.5 and for the 12 snowfall separated + bias-adjusted RCM simulations (RCM$_{sep+ba}$).

[Figure]

**Comment [Sven17]:** Figure revised due to removal of two further model chains.

**Figure S10** Relative changes (SCEN period 2070-2099 with respect to CTRL period 1981-2010) of max. 1 day snowfall, $\delta S_{1d}$, snowfall frequency, $\delta S_{freq}$, snowfall intensity, $\delta S_{freq}$, and snowfall fraction, $\delta S_{frac}$, based on the 12 snowfall separated + bias-adjusted ($RCM_{sep+ba}$) RCM simulations for RCP4.5 and RCP8.5, each. The first column shows the mean September-May snowfall index statistics vs. elevation while monthly snowfall index changes (spatially averaged over the elevation intervals <1000 m.a.s.l., 1000 m a.s.l.-2000 m a.s.l. and >2000 m a.s.l.) are displayed in columns 2-4.

---

## Author Response (AR3)

**Future snowfall in the Alps: Projections based on the EURO-CORDEX regional climate models**

Prisco Frei, Sven Kotlarski, Mark A. Liniger, Christoph Schär

**- Third response to referees and to editor -**

We thank both the anonymous reviewer and the editor for another thorough check of the revised manuscript and for taking the effort to replicate one of our reply figures. We are sorry that we could not appropriately address one remaining concern regarding the final selection of climate models. The reviewer is completely right, Figure R1 in our replies showed the summer season JJA only. We are sorry for this mistake and for not specifying the underlying time period properly in the figure caption.

Indeed, the snow (accumulation) issue in terms of an obvious feedback to the temperature change signal is most pronounced in summer. But as we already mentioned before, deficiencies especially of the two RACMO model chains (EC-EARTH-RACMO and HadGEM2-RACMO) in adjacent seasons cannot be excluded. As we show in the new Figures S11 and S12 of the revised manuscript, these deficiencies indeed remain at least for the two adjacent months May and September which are considered in our analysis. And it is only these two model chains that are obviously affected. IPSL-WRF does not seem to be problematic, but this model chain has further obvious deficiencies that in our opinion justify a removal from at least parts of the analysis. Please note that the IPSL-WRF issue became apparent in the context of the CH2018 model selection and after(!) submission of the initial manuscript. We believe that, with this additional information, it is valid to modify the set of models considered during the revision phase of a paper if properly motivated. EC-EARTH-RACMO has been removed from the analysis following a hint of the reviewer himself, for which we are thankful.

To accommodate the remaining concerns of the reviewer and the suggestions of the editor we now revised our manuscript and implemented several changes. These changes imply a consideration of potentially deficient model simulations in the analysis, but only in places where the identification of individual experiments is possible. In these cases we now consider the original ensemble of 14 GCM-RCM chains for each emission scenario (*full set*). This set comprises EC-EARTH-RACMO and IPSL-WRF. For ensemble-based analyses that do only present multi-model mean and ranges, we however employ a reduced set of 12 model chains only (*reduced set*). The composition of the two sets is described in Table 1. The motivation for removing individual simulations from the available set of all simulations is summarized in Chapter 2.2 and fully motivated (including several additional figures) in the new Supplementary Material, Part B. An additional Figure S15 is included that corresponds to the central result Figure 12 of the main manuscript but employs the full instead of the reduced model set. A comparison of both figures indicates only minor influences of the model selection on the main results and conclusions and, hence, a robust ensemble analysis that does not strongly depend on shortcomings of individual simulations. This fact is now prominently mentioned in the conclusions of the manuscript.

In addition to this, we corrected several typos and slightly modified the phrasing at a few places throughout the manuscript.

We hope that the revised version of the paper accommodates the remaining concerns and is considered as being appropriate. Please find the new version with all changes highlighted in the attachment. We are looking forward to your decision.

With kind regards,

Sven Kotlarski
(on behalf of all co-authors)

**New manuscript version with marked-up changes:**

[revised manuscript text omitted]

**Comment [Sven1]:** As requested, figure has been replaced and now shows the full model set.

[Figure]

**Figure 5** Evaluation of snowfall indices in the EVAL period 1971-2005 for the 12 snowfall separated + bias-adjusted (RCM_sep+ba) and 12 snowfall separated + not bias-adjusted (RCM_sep+nba) RCM simulations of the reduced model set vs. observation-based reference. The first column shows the mean September-May snowfall index statistics vs. elevation while the monthly snowfall indices (spatially averaged over the elevation intervals <1000 m.a.s.l., 1000 m a.s.l.-2000 m a.s.l. and >2000 m a.s.l.) are displayed in columns 2-4.

[Figure]

**Comment [Sven2]:** As requested, figure has been replaced and now shows the full model set.

**Figure 6** Spatial distribution of mean September-May snowfall, $S_{mean}$, in the EVAL period 1971-2005 and for the
12 snowfall separated + bias-adjusted RCM simulations ($RCM_{sep+ba}$) of the full model set.  Lower right
panel: observation-based reference.

[Figure]

**Comment [Sven3]:** As requested, figure has been replaced and now shows the full model set.

**Figure 7** Spatial distribution of relative changes (SCEN period 2070-2099 with respect to CTRL period 1981-2010) in mean September-May snowfall, $\delta S_{mean}$, for RCP8.5 and for the  14 snowfall separated + bias-adjusted RCM simulations ($RCM_{sep+ba}$) of the full model set. For RCP4.5, see Fig. S7.

[Figure]

**Comment [Sven4]:** As requested, figure has been replaced and now shows the full model set.

[revised manuscript text omitted]

**Comment [Sven5]:** Figure has been replaced and now shows the full model set.

**Figure S3**: Ratios (RCM simulations of the full model set divided by observational analysis) of total precipitation
sums from September to May in 1971 - 2005 vs. elevation for the Swiss domain. The linear regression line,
applied to the ratios for elevations between 250 m a.s.l. and 2750 m a.s.l., is represented by the red line.

[Figure]

**Comment [Sven6]:** Figure has been replaced and now shows the full model set.

**Figure S4**: Total winter (SEP-MAY) precipitation bias (expressed as quotient between RCM simulations of the full
model set and observations) in the EVAL period 1971-2005 for individual elevation intervals and for the full
domain (lowermost row). Left panel: Swiss domain only. Right panel: Entire Alpine analysis domain (cf. Fig. 1).
Observational reference: EOBS version 13.1 (Haylock et al., 2008) on 0.22° interpolated to the 0.11° RCM grid by
nearest neighbour interpolation.

[Figure]

Comment [Sven7]: Figure has been replaced and now shows the full model set.

**Figure S5**: As Figure S4 but for the winter (SEP-MAY) temperature bias.

[Figure]

**Figure S6** Mean September-May snowfall sum [mm] in the period 1971-2005 as represented by the 2 km
snowfall reference (*Subgrid method*; upper left), the 12 km snowfall reference on the RCM grid (*Richards method*;
upper right) and the HISTALP dataset (Chimani et al., 2011; lower).

[Figure]

Comment [Sven8]: Figure has been replaced and now shows the full model set.

**Figure S7** Spatial distribution of relative changes (SCEN period 2070-2099 with respect to CTRL period 1981-2010) in mean September-May snowfall, $\delta S_{mean}$, for RCP4.5 and for the 14 snowfall separated + bias-adjusted RCM simulations ($RCM_{sep+ba}$) of the full model set.

[Figure]

**Comment [Sven9]:** Figure has been replaced and now shows the full model set.

**Figure S8** Spatial distribution of relative changes (SCEN period 2070-2099 with respect to CTRL period 1981-2010) in heavy snowfall, $\delta S_{q99}$, for RCP4.5 and for the 12 snowfall separated + bias-adjusted RCM simulations (RCM$_{sep+ba}$) of the full model set.

[Figure]

**Comment [Sven10]:** Figure has been replaced and now shows the full model set.

**Figure S9** Spatial distribution of relative changes (SCEN period 2070-2099 with respect to CTRL period 1981-2010) in heavy snowfall, $\delta S_{q99}$, for RCP8.5 and for the 14 snowfall separated + bias-adjusted RCM simulations (RCM$_{sep+ba}$) of the full model set.

[Figure]

**Figure S10** Relative changes (SCEN period 2070-2099 with respect to CTRL period 1981-2010) of max. 1 day
snowfall, $\delta S_{1d}$, snowfall frequency, $\delta S_{freq}$, snowfall intensity, $\delta S_{freq}$, and snowfall fraction, $\delta S_{frac}$, based on the 12
snowfall separated + bias-adjusted (RCM$_{sep+ba}$) RCM simulations of the reduced model set for RCP4.5 and
RCP8.5, each. The first column shows the mean September-May snowfall index statistics vs. elevation while
monthly snowfall index changes (spatially averaged over the elevation intervals <1000 m.a.s.l., 1000 m a.s.l.-2000
m a.s.l. and >2000 m a.s.l.) are displayed in columns 2-4.

**Supplementary Material, Part B**
**Climate Model Selection**

Our analysis initially considered all EURO-CORDEX GCM-RCM combinations available in December 2016 that provide experiments for the higher EUR-11 resolution and for both RCP4.5 and RCP8.5. Out of this initial set individual combinations were either completely or partly removed from the analysis due to the reasons outlined below. The reduced model set consists of 12 GCM-RCM chains (see Table 1 of the main manuscript) and is consistent with the current model selection for the upcoming CH2018 Swiss Climate Scenarios (*www.ch2018.ch*). In the present work all ensemble-based analyses that do not allow an identification of individual experiments are carried out for this reduced set only. Fig. 12 of the main manuscript, however, is replicated for the full set which allows an intercomparison of the results for both selections (Fig. S15).

**MPI-ESM-REMO**
Two realisations of the GCM-RCM chain MPI-ESM-REMO are available (initial condition ensemble sampling internal climate variability). In order to avoid mixing GCM-RCM sampling with pure internal climate variability sampling, the second realisation of this model chain was removed and only the first one was considered.

**HadGEM2-RACMO and EC-EARTH-RACMO**
These two model chains are subject to a widespread continuous accumulation of snow cover at high alpine grid cells in the course of the 21$^{st}$ century. For individual grid cells several tens of meters of snow water equivalent are obtained. The extensive snow accumulation has obvious feedbacks on the climate change signal of 2m temperature: Temperature change signals are considerably lower at the affected grid cells than in surrounding regions. The summer season JJA, which is excluded from our analysis, is most affected (Fig. S11). But also neighbouring months are concerned (Fig. S12). The ultimate reason for this behaviour is not clear, but the issue is potentially critical for our analysis as 2m temperature change signals directly influence future snowfall changes via changes of the precipitation phase. HadGEM2-RACMO, which is subject to the highest spatial variability of temperature change signals in the May/September mean, has therefore been completely omitted in this study. EC-EARTH-RACMO is considered in the full but not in the reduced model set (see Table 1 of the main manuscript).

**IPSL-WRF**
This model chain is subject to suspicious precipitation and temperature change signals in the northern part of the Alps that at least partly have to be considered to be unphysical. The most affected season is summer (JJA), but also adjacent months are concerned. In detail, the RCM output in terms of precipitation along the northern flanks of the Alps shows a very low correlation with precipitation amounts in the driving GCM (Fig. S13) and an opposite future change signal (summer precipitation increase instead of a clear summer precipitation decrease in the driving GCM; Fig. S13). Furthermore, the simulated temperature evolution in summer (JJA) and partly also in autumn (SON) is subject to a sudden shift to lower levels around year 2023 (Fig. S14). This feature is not apparent in the driving GCM and cannot be explained on a physical basis. The IPSL-WRF model chain is therefore considered as part of the full model set, i.e. in places where identification of individual models is possible, but is not contained in the reduced model set (see Table 1 of the main manuscript).

[Figure]

**Figure S11** Spatial pattern of summer (JJA) temperature change signals (2070-2099 with respect to 1981-2010)
for RCP8.5 in the 15 EUR-11 GCM-RCM model chains that were available in December 2016. Red font denotes
models that were either completely or partly removed from our analysis. Note the considerable lower change
signal over high-alpine grid cells in EC-EARTH-RACMO and HadGEM2-RACMO.

[Figure]

**Figure S12** As Fig. S11 but for the mean May/September temperature change signal.

[Figure]

**Figure S13** Evolution of seasonal mean precipitation over north-eastern Switzerland in EUR-11 IPSL-WRF for
RCP8.5. Dark line: driving GCM (IPSL), bright line: RCM (WRF). The number in the lower right corner of each
panel indicates the non-detrended Pearson correlation coefficient of mean seasonal precipitation in the RCM and
its driving GCM. Note the negative correlation in summer (JJA) and the opposing summer precipitation trends.

[Figure]

**Figure S14** As Fig. S13 but for seasonal mean temperature. Note the low correlations of summer (JJA)
temperatures and the apparent shift of summer (JJA) and autumn (SON) temperatures in the RCM around year
2023.

[Figure]

**Figure S15** As Fig. 12 of the main manuscript but for the full model set (14 RCM simulations for each emission
scenario).

---

## Author Response (AR4)

**Future snowfall in the Alps: Projections based on the EURO-CORDEX regional climate models**

Prisco Frei, Sven Kotlarski, Mark A. Liniger, Christoph Schär

**- Final response to referees and to editor -**

Dear Editor, dear Referees,

We are happy that our revised version of the manuscript has been accepted and, once again, would like to thank you for a thorough and very helpful review of the paper. We have now uploaded the final version. With respect to the previous revised version one minor change has been introduced:

In line 181 (page 5) we changed "*and serious simulation deficiencies that potentially affect our analysis*" to "*and simulation deficiencies that potentially affect our analysis*", i.e. we removed the word "serious" from this sentence. After an additional assessment we believe that this revised phrasing better reflects the actual issue and the fact that results considering the two models removed from the full set are very similar to the results for the reduced set of models. It is also better in line with the suggestions of referee 3.

We hope this minor adjustment is considered as being appropriate.

With kind regards and, again, thanks for your work,

Sven Kotlarski
(on behalf of all co-authors)